# Learning Personalized Causally Invariant Representations for Heterogeneous Federated Clients

**Xueyang Tang[1], Song Guo[2]**[*]**, Jie Zhang[1]**[*] **& Jingcai Guo[1]**
[1]The Hong Kong Polytechnic University
[2]The Hong Kong University of Science and Technology

## Abstract

Personalized federated learning (PFL) has gained great success in tackling the scenarios where target datasets are heterogeneous across the local clients. However, the application of the existing PFL methods to real-world setting is hindered by the common assumption that the test data on each client is in-distribution (IND) with respect to its training data. Due to the bias of training dataset, the modern machine learning model prefers to rely on shortcut which can perform well on the training data but fail to generalize to the unseen test data that is out-of-distribution (OOD). This pervasive phenomenon is called shortcut learning and has attracted plentiful efforts in centralized situations. In PFL, the limited data diversity on federated clients makes mitigating shortcut and meanwhile preserving personalization knowledge rather difficult. In this paper, we analyse this challenging problem by formulating the structural causal models (SCMs) for heterogeneous federated clients. From the proposed SCMs, we derive two significant causal signatures which inspire a provable **shortcut discovery and removal** method under federated learning, namely FedSDR. Specifically, FedSDR is divided into two steps: 1) utilizing the available training data distributed among local clients to discover all the shortcut features in a collaborative manner. 2) developing the optimal personalized causally invariant predictor for each client by eliminating the discovered shortcut features. We provide theoretical analysis to prove that our method can draw complete shortcut features and produce the optimal personalized invariant predictor that can generalize to unseen OOD data on each client. The experimental results on diverse datasets validate the superiority of FedSDR over the state-of-the-art PFL methods on OOD generalization performance.

## 1 Introduction

Federated learning (FL) allows the participation of a massive number of data holders (i.e., clients) that possess limited data to collaboratively train learning models in a privacy-preserving manner (McMahan et al., 2017). From the view of the heterogeneity of target datasets across local clients, we can divide the literature on FL into two branches. 1) Federated learning aims at training a global model to fit the local data distributions and perform well when the local target datasets are subject to independent and identically distribution (IID). In particular, some works (Deng et al., 2020; Liu et al., 2021c; Nguyen et al., 2022) (including robust federated learning and federated domain generalization) focus on training a global model that can tackle the distribution/domain shift across local training datasets. Unfortunately, the shared global model can diverge from the optimal local solutions when the target datasets are heterogeneous or not IID (i.e., Non-IID) across local clients (Hsieh et al., 2020), since the useful information about personalization is dropped. 2) Personalized federated learning develops a personalized model for each client to handle the discrepancy among the local optima when the target datasets across local clients are Non-IID. Despite succeeding in handling Non-IID target datasets, all the existing PFL methods neglect the shortcut trap problem which is attracting more and more interest in centralized machine learning.

---

[*]Corresponding author

Shortcut trap is found pervasive in modern machine learning (Geirhos et al., 2020) where models prefer to rely on the shortcut to solve problems due to the bias of training dataset. The utilized shortcut can perform well on training data but fails to generalize to unseen test data that is out-of-distribution (OOD) with respect to the training data. For example, there is a binary image classification task where the model needs to recognize the pictures of cows and camels (Beery et al., 2018). Deep learning model can classify the picture of a cow in a desert background as "camel" at test time, if most of cows appear in grass backgrounds and most of camels stand in desert backgrounds in training environments (environments are data subsets that have different data distributions). This dataset bias makes the obtained model choose the background rather than the shape of animals in the pictures as the discriminative feature. The similar shortcut trap exists in diverse real-world scenarios (Geirhos et al., 2020). Although many efforts have been attracted to the shortcut trap problem in centralized situations, they focus on mitigating shortcut by extracting environment-invariant (a.k.a. invariant) features. When applying these schemes into PFL, the invariance constraint will eliminate all heterogeneous features, including shortcut and personalized features. Therefore, the existing invariant learning schemes can hardly tackle the shortcut trap problem in PFL.

What's worse, we find the trivial combination of the existing PFL and centralized invariant learning schemes, instead of solving the shortcut trap problem in PFL, can even induce worse performance than the better one of themselves (discussed in the evaluation part). To handle the challenging shortcut trap problem in PFL, we firstly formulate the structural causal models (SCMs) to simulate the heterogeneous data generating processes on local clients. From the SCMs, we derive a causal signature which reveals that the shortcut is statistical independent with the client/user indicator conditional on label and environment indicator. Inspired by this finding, we design a collaborative shortcut discovery method which can work well even if there is only one available training environment on each client. Then, the personalized causally invariant representations are extracted by utilizing another causal signature that describes the conditional independence between the personalized invariant features and the shortcut features.

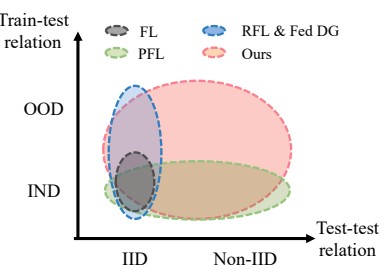

Figure 1: The coverage of ours and the related works. a) FL: **F**ederated **L**earning; b) PFL: **P**ersonalized **F**ederated **L**earning; c) RFL: **R**obust **F**ederated **L**earning; d) Fed DG: **Fed**erated **D**omain **G**eneralization. Besides, IND denotes in-distribution.

Finally, the optimal personalized invariant predictors can be elicited from the extracted personalized causally invariant features. The comparison between the coverage of our approach and the related works is illustrated in Figure 1. The main contributions of this paper are summarized as follows:

- To the best of our knowledge, we are the first to consider the shortcut trap problem in personalized federated learning and analyse it by formulating the structural causal models for heterogeneous clients. Based on the proposed SCMs, we design a provable shortcut discovery and removal method to develop the optimal personalized invariant predictor which can generalize to unseen local test distribution for each client.

- Theoretically, we demonstrate that the designed shortcut discovery method can draw all the latent shortcut components, then the shortcut removal method can eliminate the discovered shortcut features and produce the optimal personalized invariant predictor for each client.

- Empirically, we conduct experiments on several commonly used out-of-distribution datasets and the results validate the superiority of our method on out-of-distribution generalization performance, compared with the state-of-the-art competitors.

## 2 RELATED WORK

**Federated learning.** The classic FedAvg (McMahan et al., 2017) performs well if local training datasets are IID. Some methods (Karimireddy et al. (2020); Dieuleveut et al. (2021); Zhang et al. (2022)) mitigate the negative impact of training data heterogeneity on convergence rate, while another branch ( Deng et al. (2020); Sharma et al. (2022); Sun & Wei (2022)) targets at reducing the performance bias of global model on local clients. Besides, few works ( Liu et al. (2021c); Nguyen et al. (2022); Guo et al. (2023c)) investigate the scenarios where the training data heterogeneity ap-

pears to be domain shift. All the above methods produce a shared global model which can diverge from the local optimal solutions when local target datasets are Non-IID.

**Personalized federated learning.** Many PFLs ( T Dinh et al. (2020); Hanzely et al. (2020); Fallah et al. (2020); Li et al. (2021); Tang et al. (2022); Cheng et al. (2023)) train the personalized models with the guidance of a global model which embeds in the shared knowledge, DFL (Luo et al., 2022) disentangles the shared features from the client-specific ones to achieve accurate aggregation on shared knowledge. Similarly, pFedPara (Hyeon-Woo et al., 2022) and Factorized-FL (Jeong & Hwang, 2022) factorize the model parameters into the shared and personalized parts. Another branch ( Collins et al. (2021); Chen & Chao (2022); Xu et al. (2023)) employs the shared/aligned feature extractor to capture global knowledge and personalized classifiers to encode the personalization information. All of them don't cover the situations where shortcut exists in local training datasets.

**Shortcut and Invariant learning (IL).** Causally invariant predictor is proposed in (Peters et al., 2016), and then applied into deep learning in IRM (Arjovsky et al., 2019) to mitigate shortcut. Subsequently, Rosenfeld et al. (2021) prove that IRM and its variants can be still trapped by shortcut when training environments are insufficient. IFM (Chen et al., 2022b) lowers the requirement and demands only logarithmic training environments. Some works focus on settling IL problem when the environment label is unavailable, e.g., EIIL (Creager et al., 2021), HRM (Liu et al., 2021a;b), EDNIL (Huang et al., 2022) and ZIN (Lin et al., 2022). Another branch (Ahuja et al., 2021); Chen et al. (2022a); Huh & Baidya (2022)) completes the constraints that IRM misses to improve the performance. The iCaRL (Lu et al., 2022) extends IL to non-linear causal representations while ACTIR (Jiang & Veitch, 2022) extends IL to anti-causal scenarios. All these methods are devised for centralized scenarios where all training data is accessed and training environments are sufficient.

## 3 PROBLEM FORMULATION

**Notations.** Let $\mathcal{X}$, $\mathcal{Y}$ and $\mathcal{E}$ denote the input, target and environment space respectively. Data instance is $(X, y, e) \in (\mathcal{X}, \mathcal{Y}, \mathcal{E})$. Suppose there are $N$ clients and the local dataset $D_u$ on client $u$ contains $M_u$ samples, $u \in [N]$. The sets of training and test environments on client $u$ are denoted by $\mathcal{E}_{tr}^u$ and $\mathcal{E}_{te}^u$ respectively. We use $\mathcal{E}_{all}^u$ as the set of all possible environments in the task that client $u$ concentrates on, i.e., $\mathcal{E}_{tr}^u, \mathcal{E}_{te}^u \subset \mathcal{E}_{all}^u, \forall u \in [N]$. In federated learning system, the overall environment sets are denoted by $\mathcal{E}_{tr} := \bigcup_u \mathcal{E}_{tr}^u$ and $\mathcal{E}_{all} := \bigcup_u \mathcal{E}_{all}^u$. For convenience, we separate the learning model or parameterized mapping from $\mathcal{X}$ to $\mathcal{Y}$ into two consecutive parts: **1)** the feature extractor ($\Phi$ and $\Psi$ denote the invariant and spurious feature extractors respectively) maps from input space $\mathcal{X}$ to latent feature space $\mathcal{Z}$, i.e., $\Phi(X) \in \mathcal{Z}$ and $\Psi(X) \in \mathcal{Z}$; **2)** the classifier $\omega$ outputs a prediction $\hat{y}$ from a latent feature $z \in \mathcal{Z}$. For example, the overall model based on the invariant feature extractor is denoted by $f_\theta(\cdot) = \omega(\Phi(\cdot))$ where $f_\theta$ indicates the function $f$ parameterized by $\theta$. We define the expected empirical loss for model $f_\theta$ on dataset $D$ as $\mathcal{R}(f_\theta; D) := \mathbb{E}_{(X,y) \in D}[\ell(f_\theta(X), y)]$ where $\ell$ is the loss function.

### 3.1 INVARIANT LEARNING

Succeeding in mitigating shortcut and solving the OOD generalization problem, invariant learning assumes that there exists some invariant feature $\Phi(X)$ satisfying the ***invariance constraint***:

$$\mathbb{P}(Y|\Phi(X) = z, e) = \mathbb{P}(Y|\Phi(X) = z, e'), \forall z \in \mathcal{Z}, \forall e, e' \in \mathcal{E}_{all}. \tag{1}$$

Rosenfeld et al. (2021) proved that IRM (Arjovsky et al., 2019) and its variants need at least $d_S + 1$ ($d_S$ is the dimension of shortcut features) training environments to eliminate all shortcut features and elicit the optimal invariant predictor, under linear scenarios.

**Definition 1 (Optimal Invariant Predictor)** *The optimal invariant predictor is elicited based on the complete invariant features which are informative for the target label in the task concerned, i.e.,* $\Phi^\star = \arg\max_\Phi I(Y; \Phi(X))$, *where* $I(\cdot; \cdot)$ *denotes the Shannon mutual information between two random variables and $\Phi$ satisfies the above invariance constraint.*

### 3.2 CAUSAL SETUP

In invariant learning (IL), researchers usually formulate a structural causal model to simulate the data generating process in the target task. A valid SCM is depicted by a directed acyclic graph where each

node represents a random variable and each edge describe a directed functional relationship between the corresponding variables (Pearl, 2009). When we study the invariant learning in federated setting, the latent heterogeneity of data generating mechanisms among local clients need to be considered.

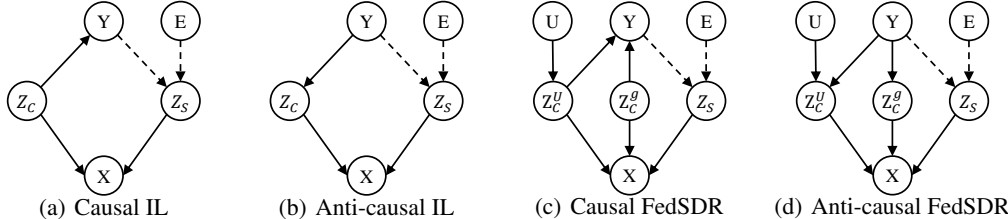

(a) Causal IL      (b) Anti-causal IL      (c) Causal FedSDR      (d) Anti-causal FedSDR

Figure 2: (a) (Arjovsky et al., 2019; Huang et al., 2022) and (b) (Rosenfeld et al., 2021; Huh & Baidya, 2022) give the structural causal models (SCMs) commonly adopted in invariant learning, while (c) and (d) show the SCMs proposed in this paper. $Z_C$ and $Z_S$ denote the invariant and shortcut features respectively. $E$ is the indicator of shortcut while $U$ is the indicator of user/client. Dotted arrows indicate unstable causal relations that can vary in different environments.

Therefore, we propose the SCMs in federated learning by adding the **u**ser/client indicator $U$ and deconstructing the invariant features into two separate parts: the personalized invariance $Z_C^U$ and the shared/global invariance $Z_C^g$. The detailed SCMs are shown in Figure 2. As discussed in the literature on invariant learning, $Z_S$ is the latent shortcut feature. The functional relation between $Z_S$ and label $Y$ can vary across different environments. That is, $\forall Z_S$ there always exists some $e, e' \in \mathcal{E}_{all}$ that make $\mathbb{P}(Y|Z_S, e) \neq \mathbb{P}(Y|Z_S, e')$ hold. By analogy with the optimal invariant predictor in invariant learning, we provide the definition of the optimal personalized invariant predictor in PFL.

**Definition 2 (Optimal Personalized Invariant Predictor)** *The optimal personalized invariant predictor for client $u$ is elicited based on the complete invariant features which are informative for target label in the task that client $u$ concentrates on. That is, $\Phi_u^\star = \arg\max_{\Phi_u} I(Y; \Phi_u(X))$, where $\Phi_u$ satisfies that $\mathbb{P}(Y|\Phi_u(X) = z, e) = \mathbb{P}(Y|\Phi_u(X) = z, e'), \forall z \in \mathcal{Z}, \forall e, e' \in \mathcal{E}_{all}^u$.*

## 4 METHOD: FEDSDR

When the training environments on each client are insufficient, locally invariant learning can fail as discussed before. *How about a collaborative manner?* Unfortunately, personalized invariant features can cause deviation of the invariance constraint as shortcut features do if training environments are collected from different clients. As a result, the collaborative invariant learning can eliminate/preserve both personalized invariant and shortcut features with the same probability. *Nonetheless, how about combining collaborative invariant learning with PFL methods?* Even though we can get the global invariant features via collaborative IL and conduct local adaptation as in many PFL schemes (e.g., fine-tuning (Cheng et al., 2023) and L2-regularization (T Dinh et al., 2020; Hanzely et al., 2020; Li et al., 2021)), the local adaptation can pick up both personalized invariant and shortcut features again since local training environments are insufficient. It turns out the trivial combination can hardly outperform the superior individual one on OOD generalization performance.

**FedSDR.** In view of the above failure, we turn to the complementary perspective: discovering the shortcut features and removing them instead of straightly constraining invariance. The feasibility of this tack is guaranteed by the causal signatures that we derive from the SCMs in Figure 2(c) and 2(d).

**Lemma 1** *If the data generating mechanism of each federated client obeys the causal graph in Figure 2(c) or the anti-causal graph in Figure 2(d), we can have:*

- *$Z_S \perp\!\!\!\perp U \mid Y, E$ which means that the shortcut features $Z_S$ are conditionally independent of the personalization indicator $U$ given $Y$ and $E$.*

- *$Z_C^g \perp\!\!\!\perp Z_S \mid Y$ and $Z_C^U \perp\!\!\!\perp Z_S \mid Y$, which means that both the global ($Z_C^g$) and personalized ($Z_C^U$) invariant features are conditionally independent of the shortcut features $Z_S$ given $Y$.*

**Remark 1.** The first causal signature in Lemma 1 indicates that we can discover the shortcut features using training environments across local clients even if the data generating mechanisms are

heterogeneous among them. The second causal signature makes it possible to develop the optimal personalized invariant predictors with the discovered shortcut features even though there is just one training environment on each client, since the relationships between $Z_C^g$, $Z_C^U$ and $Z_S$ are independent of environment $E$. The detailed proof of Lemma 1 can be found in the appendix.

In the following sections, we will introduce the two-stage implementation of our method in detail.

## 4.1 THE PROVABLE SHORTCUT DISCOVERY

At the first stage, we need to capture the complete shortcut features in a collaborative manner. Recalling the difference between the definitions of shortcut features and invariant features, we design the following objective to extract the complete shortcut features in a collaborative manner:

$$\omega_\Psi^\star, \Psi^\star = \underset{\substack{\Psi:\mathcal{X}\to\mathcal{H}\\ \omega:\mathcal{H}\to\mathcal{Y}}}{\arg\min} \frac{1}{N}\sum_{u=1}^N \{\ell_{SD}^u(\Psi; D_u) := \mathcal{R}(\omega(\Psi); D_u) - \lambda\,\ell_{dis}(\Psi; D_u)\}, \tag{2}$$

where the first term $\mathcal{R}(\omega(\Psi); D_u)$ is adopted to exclude the uninformative features (e.g., noise). $\lambda$ is the balancing weight and the second term $\ell_{dis}(\Psi; D_u)$ is designed for extracting the complete shortcut features. Specifically, we define that

$$\ell_{dis}(\Psi, D_u) := \mathbb{E}_{X\in D_u}\Big[\sum_{e_i\in\mathcal{E}_{tr}}\sum_{e_j\in\mathcal{E}_{tr}}\mathcal{KL}\big(\mathbb{P}_{\omega_i^\star}(Y\mid\Psi(X),e_i)\big\|\mathbb{P}_{\omega_j^\star}(Y\mid\Psi(X),e_j)\big)\Big], \tag{3}$$

where $\mathcal{KL}(\mathbb{P}\|\mathbb{Q})$ denotes the Kullback–Leibler divergence between two probability distributions. $\mathbb{P}_{\omega_i^\star}(Y\mid\Psi,e_i)$ means that $\mathbb{P}(Y\mid\Psi,e_i)$ is parameterized by the classifier $\omega_i^\star$ which is trained by:

$$\omega_i^\star = \underset{\omega_i:\mathcal{H}\to\mathcal{Y}}{\arg\min}\sum_{u=1}^N \rho_u^i\mathcal{R}(\omega_i(\Psi); e_i), \forall e_i\in\mathcal{E}_{tr}, \tag{4}$$

where $\rho_u^i = 1$ when client $u$ has data samples from environment $e_i$ and $\rho_u^i = 0$ otherwise.

Since $\mathbb{P}_{\omega_i^\star}(Y\mid\Psi,e_i)$ is parameterized by the classifier $\omega_i^\star$ to be a distribution around $\omega_i^\star(\Psi)$ for any given $\Psi$ and $e_i$, we adopt a simple and effective measure to compute the divergence $\mathcal{KL}(\mathbb{P}_{\omega_i^\star}(Y\mid\Psi,e_i)\|\mathbb{P}_{\omega_j^\star}(Y\mid\Psi,e_j))$. That is $\mathcal{KL}(\mathbb{P}_{\omega_i^\star}(Y\mid\Psi,e_i)\|\mathbb{P}_{\omega_j^\star}(Y\mid\Psi,e_j)) = \frac{1}{2}\|\omega_i^\star(\Psi) - \omega_j^\star(\Psi)\|^2$, $\forall e_i, e_j\in\mathcal{E}_{all}$. In this way, we can rewrite the overall objective 2 for shortcut discovery as the following bi-level optimization:

$$\omega_\Psi^\star, \Psi^\star = \underset{\Psi,\omega}{\arg\min} \frac{1}{N}\sum_{u=1}^N \{\ell_{SD}^u(\Psi; D_u) := \mathcal{R}(\omega(\Psi); D_u) - \lambda\,\ell_{dis}(\Psi; D_u)\} \tag{5}$$

$$\text{s.t.}\qquad \omega_i^\star = \underset{\omega_i:\mathcal{H}\to\mathcal{Y}}{\arg\min}\sum_{u=1}^N \rho_u^i\mathcal{R}(\omega_i(\Psi); e_i), \forall e_i\in\mathcal{E}_{tr}, \tag{6}$$

where $\ell_{dis}(\Psi; D_u) = \mathbb{E}_{X\in D_u}[\frac{1}{2}\sum_{e_i\in\mathcal{E}_{tr}}\sum_{e_j\in\mathcal{E}_{tr}}\|\omega_i^\star(\Psi(X)) - \omega_j^\star(\Psi(X))\|^2]$. This bi-level optimization can be solved by alternatively updating the solutions of the outer and inner objective. Under federated learning, both the outer and inner objective can be divided into $N$ sub-problems that can be settled on $N$ local clients respectively. The sever can aggregate the update from local clients to gain the solution $\Psi^\star$. To avoid the outer objective being dominated by maximizing $\ell_{dis}(\Psi; D_u)$, we replace $\ell_{dis}$ with $\min(\alpha, \lambda\,\ell_{dis}(\Psi; D_u))$ in the practical version, where $\alpha$ is a positive threshold.

**Theoretical Analysis.** Before continuing to introduce the shortcut removal method, we formally analyse the optimal solution of the Eq. 5. In this theoretical analysis part, we consider the linear data model that Rosenfeld et al. (2021) adopted. Specifically, we focus on the logistic regression problem where label $y \in \{\pm 1\}$. As in (Rosenfeld et al., 2021), we suppose both the invariant features $Z_C = [Z_C^g, Z_C^U]$ and shortcut features $Z_S$ of sample $y$ are drawn from the following Gaussian:

$$Z_C \sim \mathcal{N}(y\cdot\mu_c, \sigma_c^2 I), \qquad Z_S \sim \mathcal{N}(y\cdot\mu_s, \sigma_s^2 I),$$

where $\mu_c \in \mathbb{R}^{d_c}$ and $\mu_s \in \mathbb{R}^{d_s}$. The sample of observation $X$ is generated by $X = g(Z_C, Z_S)$ where $g(\cdot)$ is a non-parameterized function. The parameters $\mu_c$, $\sigma_c$ and function $g$ are independent of environment, while $\mu_s$ and $\sigma_s$ varies in different environments.

**Assumption 1** *The distribution of label $Y$ satisfies the following two conditions: 1) $\mathbb{P}(Y\mid u) = \mathbb{P}(Y), \forall u\in U$; 2) $\mathbb{P}(y\mid e) = \mathbb{P}(y'\mid e), \forall y, y'\in Y$ and $\forall e\in\mathcal{E}_{tr}$.*

**Theorem 1** *If the Assumption 1 holds, the function $g$ is linear and the total number of training environments in the federated learning system satisfies $|\mathcal{E}_{tr}| > d_s$, then the following two statements are equivalent:*

- $\Psi^\star(X)$ *depends and only depends on the complete shortcut features $Z_S$. That is, $\Psi^\star(X)$ is a function of $Z_S$ alone;*

- $\Psi^\star$ *is the optima of the Eq. 5 with an appropriately chosen hyper-parameter $\lambda$.*

**Remark 2.** Theorem 1 guarantees the elaborated Eq. 5 can yield the feature extractor that extracts complete shortcut features and excludes all invariant features. Note that Assumption 1 is about the label distributions in training datasets. Since the shortcut extractor works as an auxiliary model and is never part of the optimal personalized invariant predictors, we can sample some data subsets from local training datasets to train the shortcut extractor $\Psi^\star$. In this way, the sampled data subsets can easily satisfy Assumption 1. Besides, the causal signatures in Lemma 1 play critical parts in the proof of Theorem 1 and the complete proof is provided in the appendix.

## 4.2 Personalized Invariant Learning with Shortcut Removal

With the shortcut extractor that depends and only depends on the complete shortcut features $Z_S$, we can extract the most informative invariant features to elicit the optimal personalized invariant predictor for each client. Based on the second causal signature in Lemma 1, we design the following objective for each client to develop the optimal personalized invariant predictor:

$$\omega_u^\star(\Phi_u^\star) = \underset{\Phi_u, \omega_u}{\arg\min}\, \ell_{SR}^u(\omega_u(\Phi_u); D_u) := \{\mathcal{R}(\omega_u(\Phi_u); D_u) + \gamma \cdot I(\Phi_u; \Psi^\star \mid Y)\}, \forall u \in [N], \quad (7)$$

where $I(\cdot; \cdot \mid \cdot)$ denotes the conditional mutual information and $\gamma$ is the balancing weight. The optimal personalized invariant predictor is given by $f_{\theta_u}^\star := \omega_u^\star(\Phi_u^\star)$.

**Theorem 2** *Suppose $\Psi^\star(X)$ in the Eq. 7 depends and only depends on the complete shortcut features $Z_S$. If $f_{\theta_u}^\star (\forall u \in [N])$ is the optima of the Eq. 7 with the hyper-parameter $\gamma$ chosen appropriately, then the $f_{\theta_u}^\star$ is the optimal personalized invariant predictor for the client $u$, $\forall u \in [N]$.*

**Remark 3.** Theorem 2 guarantees that our method can produce the optimal personalized invariant predictor for every client. Note that $I(\Phi_u; \Psi^\star \mid Y) = 0$ is the necessary and sufficient condition for $\Phi_u \perp\!\!\!\perp \Psi^\star \mid Y$. Since $\Phi_u \perp\!\!\!\perp \Psi^\star \mid Y$ is independent of environment, our method can develop the optimal personalized invariant predictor for every client even though there is only one training environment on each client. The complete proof of Theorem 2 is provided in the appendix.

In the practical implementation, it can be infeasible to compute the exact value of $I(\Phi_u; \Psi^\star \mid Y)$. Considering the limited computation resources on local clients, we adopt a simple approximating scheme used in (Jiang & Veitch, 2022) to measure $I(\Phi_u; \Psi^\star \mid Y)$. Specifically, we estimate it by $I(\Phi_u; \Psi^\star \mid Y) \approx \mathbb{E}[\Phi_u(X) \cdot (\Psi^\star(X) - \mathbb{E}[\Psi^\star(X) \mid Y])]$ because $I(\Phi_u; \Psi^\star \mid Y) = 0$ is the sufficient (but not necessary) condition for $\mathbb{E}[\Phi_u(X) \cdot (\Psi^\star(X) - \mathbb{E}[\Psi^\star(X) \mid Y])] = 0$. With the data samples on local clients, we estimate the conditional mutual information by

$$I(\Phi_u; \Psi^\star \mid Y) \approx \left\| \frac{1}{M_u} \sum_{m=1}^{M_u} \Phi_u(X_m) \Big( \Psi^\star(X_m) - \sum_{n=1}^{M_u} \frac{q_n^m}{\sum_{n \in [M_u]} q_n^m} \Psi^\star(X_n) \Big) \right\|_1$$

where $(X_m, y_m), m \in [M_u]$ is drawn from dataset $D_u$, $q_n^m = 1$ if $y_n = y_m$ and $q_n^m = 0$ otherwise.

Note that our method can easily cooperate with most of the existing PFL methods to improve their OOD generalization performance by adding $I(\Phi_u; \Psi^\star \mid Y)$ into their objectives as a regularization term, since $I(\Phi_u; \Psi^\star \mid Y) = 0$ can constrain the personalized models to eliminate all shortcut features even though each client has only one training environment.

## 4.3 Algorithm Design

In the following contents, we will discuss what the server and local clients need to conduct to develop the optimal personalized invariant predictor $f_{\theta_u}^\star$ for each client, $\forall u \in [N]$.

**Server Update.** Before the algorithm starts, the server initializes the models with random parameters. At each communication round $t$, the server firstly selects a fraction of local clients ($u \in S^t$)

and broadcast the current $\Psi^t$ and $\{\omega_i^t \mid i = 1, 2, ..., |\mathcal{E}_{tr}|\}$ to them. After the selected local clients finish conducting the **client update** process, the server can receive the local update $\Psi_u^{t+1}$ and $\{\omega_{i,u}^{t+1} \mid i = 1, 2, ..., |\mathcal{E}_{tr}^u|\}$ from the selected clients. Then it can update the global solutions by $\Psi^{t+1} = \frac{1}{|S^t|} \sum_{u \in S^t} \Psi_u^{t+1}$ and $\omega_i^{t+1} = \sum_{u \in S^t} \frac{\rho_u^i}{\sum_{u \in S^t} \rho_u^i} \omega_{i,u}^{t+1}, i = 1, 2, ..., |\mathcal{E}_{tr}|$.

**Client Update.** Before the algorithm starts, the client $u$ initializes the personalized invariant model with random parameters $f_{\theta_u}^0$. After receiving the global model $\Psi^t$ and $\{\omega_i^t \mid i = 1, 2, ..., |\mathcal{E}_{tr}|\}$ from the server, the local client $u$ ($\forall u \in S^t$) needs to carry on the following two steps: **1)** update the personalized invariant model by

$$f_{\theta_u}^{t,k+1} = f_{\theta_u}^{t,k} - \eta \nabla \ell_{SR}^u(f_{\theta_u}^{t,k}; D_u)$$

for $K$ steps and finally get $f_{\theta_u}^{t+1} = f_{\theta_u}^{t,K}$, where $\eta$ is the personalized learning rate. **2)** The client can conduct $R$ local iterations to update the local shortcut extractor. Before it starts, the client initializes the related models as $\Psi_u^{t,r=0} = \Psi^t$ and $\omega_{i,u}^{t,r=0} = \omega_i^t, i = 1, 2, ..., |\mathcal{E}_{tr}^u|$. During each local iteration $r$, the client firstly updates the local shortcut extractor by

$$\Psi_u^{t,r+1} = \Psi_u^{t,r} - \beta \nabla \ell_{SD}^u(\Psi_u^{t,r}; D_u)$$

for one epoch where $\beta$ denotes the learning rate, and then get the near-optimal environment classifiers $\omega_{i,u}^{t,r+1}, i = 1, 2, ..., |\mathcal{E}_{tr}^u|$ by stochastic gradient descent (on $\nabla \mathcal{R}(\omega_{i,u}^{t,r}(\Psi_u^{t,r}); e_i)$) for $L$ steps. When completing $R$ local iterations, the client upload the local parameters $\Psi_u^{t+1} = \Psi_u^{t,R}$ and $\{\omega_{i,u}^{t+1} = \omega_{i,u}^{t,R} \mid i = 1, 2, ..., |\mathcal{E}_{tr}^u|\}$ to the server for **server update**.

Due to the space limitation, we place the pseudo-code of the above algorithm and more discussions on the combinations of our method with the prevalent PFL schemes in the appendix.

## 5 Experiments (More details and results in the appendix)

### 5.1 Experimental Setup

**Colored-MNIST (CMNIST)** (Arjovsky et al., 2019) is constructed based on MNIST (LeCun et al., 1998) via rearranging the images of digit 0-4 into a single class labeled 0 and the images of digit 5-9 into another class labeled 1. Each digit having label 0 is colored green/red with probability $p^e/1-p^e$ and each digit having label 1 is colored red/green with probability $p^e/1 - p^e$, respectively. Thus "color" feature is shortcut in the dataset and the data distribution varies as $p^e$ changes. We provide two training environments ($p_{tr}^e = 0.90$ and $0.80$) as $\mathcal{E}_{tr}$ and every local client only has one training environment which is randomly sampled from $\mathcal{E}_{tr}$. To assess the model performance on different test distributions, the test environment on each client varies from $p_{te}^e = 0.00$ to $1.00$. Considering the heterogeneous data generating process across local clients, the data instances used for constructing the training/test environments on each client are randomly sampled from only two digit sub-classes labeled 0 (e.g., digit 1, 2) and two digit sub-classes labeled 1 (e.g., digit 6, 7) without replacement.

**Colored Fashion-MNIST (CFMNIST)** (Ahuja et al., 2020) is constructed using the same strategy as Colored-MNIST, but the original images come from Fashion-MNIST (Xiao et al., 2017). Hence, CFMNIST dataset carries more complex feature space than colored-MNIST does.

**WaterBird** (Sagawa et al., 2019) considers a real-world scenario where the photographs of waterbirds usually have water backgrounds while the photographs of landbirds usually have land backgrounds because of the distinct habitats. It makes learning models easily trapped by "background" shortcut when classify "waterbird" and "landbird". In WaterBird, a waterbird is placed onto a water/land background with probability $p^e/1-p^e$ and a landbird is placed onto a land/water background with probability $p^e/1 - p^e$ respectively. We setup two training environments ($p_{tr}^e = 0.95$ and $0.85$) as $\mathcal{E}_{tr}$ and each client has only one training environment which is randomly sampled from $\mathcal{E}_{tr}$. The test environment varies from $p_{te}^e = 0.00$ to $1.00$. We notice that the diverse geographic distributions of different bird species naturally accord with the heterogeneity of local data generating process if the federated clients are located in different geographic areas. Considering WaterBird includes 46 waterbird species and 154 landbird species, we distribute 15 (10 separated and 5 overlapped) waterbird species and 51 (34 separated and 17 overlapped) landbird species to each client. The training and test datasets on each client contain bird pictures that belong to the same bird species.

**PACS** (Li et al., 2017) is a larger real-world dataset commonly-used in evaluating out-of-distribution (OOD) generalization. It consists of 7 classes distributed across 4 environments (or domains). We adopt the "leave-one-domain-out" strategy to evaluate the OOD generalization performance. Taking personalization into consideration, we split each training domain into two subsets according to classes (i.e., one subset consists of dog, elephant and giraffe and another subset consists of guitar, horse, house, and person), and then distribute these two subsets onto two clients respectively. The training and test datasets on each client come from different domains but consist of the same classes.

**Model Selection and Competitors.** For CMNIST and CFMNIST, we adopt the deep neural network with one hidden layer as feature extractor and an subsequent fully-connected layer as classifier. As regard to Waterbird and PACS, ResNet-18 He et al. (2016) is used as the learning model where the part before the last fully-connected layer works as feature extractor and the last fully-connected layer works as classifier. We compare our method (FedSDR) with 10 state-of-the-art algorithms: 4 federated learning methods (FedAvg (McMahan et al., 2017), DRFA (Deng et al., 2020), FedSR (Nguyen et al., 2022) and FedIIR (Guo et al., 2023c)), and 6 personalized federated learning methods (pFedMe (T Dinh et al., 2020), Ditto (Li et al., 2021), FTFA (Cheng et al., 2023), FedRep (Collins et al., 2021), FedRoD (Chen & Chao, 2022) and FedPAC (Xu et al., 2023)).

Table 1: The overall comparison between the performance of our method and the baselines on four datasets.

| Dataset | CMNIST | | CFMNIST | | WaterBird | | PACS | |
|---|---|---|---|---|---|---|---|---|
| Test accuracy (%) | worst-case | average | worst-case | average | worst-case | average | worst-case | average |
| FedAvg | 3.39 | 51.03 | 0.16 | 50.02 | 54.13 | 67.95 | 41.71 | 47.66 |
| DRFA | 21.15 | 52.81 | 19.84 | _53.88_ | 59.75 | 68.39 | 42.48 | 48.95 |
| FedSR | 46.93 | 48.62 | 47.61 | 48.90 | _61.75_ | _71.68_ | 46.76 | 51.25 |
| FedIIR | _47.25_ | 48.39 | _48.06_ | 49.16 | 61.24 | 70.87 | 47.03 | 51.58 |
| FTFA | 15.42 | _54.96_ | 11.35 | 53.52 | 54.38 | 69.68 | 40.89 | 48.79 |
| pFedMe | 21.30 | 48.53 | 4.22 | 51.26 | 55.63 | 68.24 | 45.24 | 51.33 |
| Ditto | 3.02 | 50.97 | 0.37 | 50.12 | 53.13 | 68.73 | 44.95 | 51.28 |
| FedRep | 2.76 | 50.83 | 0.11 | 50.01 | 52.88 | 70.23 | 49.27 | 53.75 |
| FedRoD | 9.09 | 50.84 | 1.23 | 51.57 | 52.36 | 70.86 | 48.16 | 52.92 |
| FedPAC | 1.01 | 50.05 | 0.16 | 50.13 | 45.08 | 65.57 | _49.93_ | _54.20_ |
| **FedSDR** | **53.88** | **55.59** | **56.92** | **61.88** | **65.25** | **73.20** | **52.14** | **56.18** |

## 5.2 EXPERIMENTAL RESULTS

**Performance Comparison.** We summarize the test accuracy of all competitors on different unseen test distributions (11 test distributions in CMNIST, CFMNIST and WaterBird; 4 test distributions in PACS) and figure out the worst-case and average accuracy of each method in Table 1. We can find that our method FedSDR consistently outperform the baselines on both worst-case and average test accuracy. In particular, FedSDR achieves around **6.5**%, **9**%, **3.5**% and **2**% higher worst-case accuracy than the second best algorithm and in the meanwhile reaches the highest average accuracy on CMNIST, CFMNIST, WaterBird and PACS, respectively.

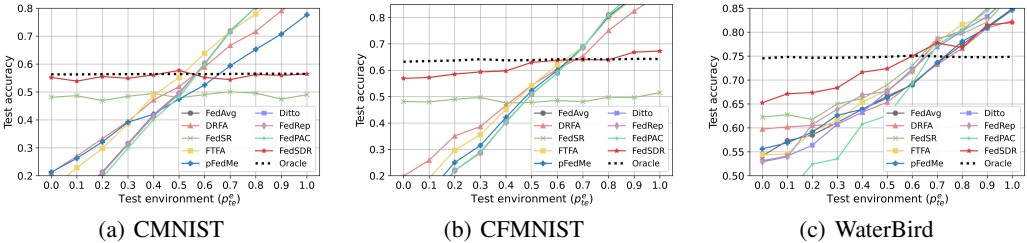

| (a) CMNIST | (b) CFMNIST | (c) WaterBird |
|---|---|---|

Figure 3: The relationship between the test accuracy and the test distribution.

**Mitigation of Shortcut Features.** Since there exists definite correlation between shortcut features and label in CMNIST, CFMNIST and WaterBird, we can use these three datasets to evaluate how well a method can mitigate the shortcut features. The more highly a method relies on shortcut, the more approximate its test accuracy is to the corresponding $p_{te}^e$. In contrast, a method that eliminates the shortcut can produce consistent test accuracy across different $p_{te}^e$. We evaluate the competitors

under diverse test environment (i.e., $p_{te}^e$) and show the relationships between test accuracy and $p_{te}^e$ in Figure 3. In particular, "Oracle" represents the scheme where we manually remove the shortcut features (color in CMNIST and CFMNIST; background in WaterBird) from the whole dataset and then train the personalized models using the pre-processed dataset. Hence "Oracle" provides an ideal performance for comparison. We can see that FedSDR can effectively mitigate the shortcut and achieve a more consistent test accuracy than most of the FL and PFL methods. Because FedSDR exploits the personalized invariant features, it consistently achieves a higher test accuracy than the federated domain generalization methods which drop the personalization information.

Table 2: Performance comparison between FedSDR and the trivial combination of IL with PFL schemes.

| Dataset | CMNIST | | CFMNIST | | WaterBird | | PACS | |
|---|---|---|---|---|---|---|---|---|
| Test accuracy (%) | worst-case | average | worst-case | average | worst-case | average | worst-case | average |
| IRM$^\dagger$ | 46.38 | 49.14 | 47.76 | 49.41 | 60.38 | 68.63 | 46.35 | 50.83 |
| IRM$^\dagger$-FT | 14.32 | 54.27 | 11.09 | 53.48 | 60.25 | 69.46 | 43.18 | 50.04 |
| IRM$^\dagger$-L2 | 45.68 | 49.04 | 47.92 | 49.46 | 61.25 | 68.93 | 48.57 | 51.98 |
| **FedSDR** | **53.88** | **55.59** | **56.92** | **61.88** | **65.25** | **73.20** | **52.14** | **56.18** |

**Necessity of Shortcut Discovery and Removal.** At the beginning of Section 4, we analyse that trivial combination of invariant learning scheme with local adaptation (commonly used in PFL) can fail to generate the optimal personalized invariant predictors for local clients. To validate the superiority of our method on developing the personalized invariant predictors when local training environments are insufficient, we implement two typical personalization skills with the global model being trained by the distributional version of IRM (i.e., IRM$^\dagger$ in Table 2). One is L2-norm regularizer used in PFL (T Dinh et al., 2020; Hanzely & Richtárik, 2020; Hanzely et al., 2020; Li et al., 2021), and we call this implementation IRM$^\dagger$-L2. Another one is local **F**ine-**T**uning which is proved simple and effective for personalization ( Cheng et al. (2023)) and we name it IRM$^\dagger$-FT.

From the results in Table 2, we can find the combinations can hardly improve the OOD generalization performance. In particular, the local fine-tuning skill can even degrade the performance, compared with baseline IRM$^\dagger$. The underlying reason is that local adaptation can readily make the personalized model pick up the shortcut features when local training environments are insufficient. By contrast, our shortcut removal method is independent of environment and can effectively mitigate the shortcut features even though there is only one training environment on each client.

**Effect of Hyper-parameter $\lambda$ and $\gamma$.** We evaluate the effects of two significant hyper-parameters in the proposed objective (i.e., $\lambda$ and $\gamma$) on model performance here. Since the results on other datasets present the similar tendency as on WaterBird, we herein focus on WaterBird. The results on other datasets are placed in the appendix. When evaluating the effect of $\lambda$, we fix $\gamma = 1.4$ . When evaluating the effect of $\gamma$, we fix $\lambda = 0.5$. The results are shown in Table 3. When $\lambda = 0.0$, shortcut feature

Table 3: Performance of FedSDR on WaterBird with different values of hyper-parameters $\lambda$ and $\gamma$.

| $\lambda$ | 0.00 | 0.10 | 0.50 | 1.00 | 10.0 |
|---|---|---|---|---|---|
| worst-case (%) | 61.88 | 62.51 | 65.25 | 61.68 | 61.74 |
| average (%) | 71.34 | 72.39 | 73.20 | 70.61 | 70.18 |

| $\gamma$ | 0.00 | 0.10 | 1.00 | 1.40 | 10.0 |
|---|---|---|---|---|---|
| worst-case (%) | 43.75 | 44.16 | 57.64 | 65.25 | 48.29 |
| average (%) | 66.30 | 65.73 | 70.18 | 73.20 | 64.86 |

extractor is trained by empirical risk minimization (i.e., ERM). When $\gamma = 0.0$, the personalized models are trained by local ERM. Because models trained by ERM tend to rely on shortcut, the performance of FedSDR is more sensitive to the selection of $\gamma$ than the selection of $\lambda$.

## 6 CONCLUSION AND FUTURE WORK

In this paper, we study the challenging shortcut trap problem in PFL. We formulate the SCMs to interpret the heterogeneous data generating mechanisms on federated clients and derive two significant causal signatures which inspire our provable shortcut discovery and removal method. Theoretical analysis proves the proposed FedSDR can draw all shortcut features and elicit the optimal personalized invariant predictor that can generalize to unseen target data for each client. FedSDR can cooperate with most of the existing PFL methods to improve their OOD generalization performance, which can facilitate the real-world application of PFL. Since the theoretical guarantee is derived under linear cases in this paper, we will extend it to more complex cases in the future work.

## ACKNOWLEDGMENTS

This research was supported by fundings from the Key-Area Research and Development Program of Guangdong Province (No. 2021B0101400003), Hong Kong RGC Research Impact Fund (No. R5060-19, No. R5034-18), Areas of Excellence Scheme (AoE/E-601/22-R), General Research Fund (No. 152203/20E, 152244/21E, 152169/22E, 152228/23E).

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

## A  RELATED WORK

Due to the space limitation, a more complete literature review is placed in this section.

**Federated learning.** The classic FedAvg (McMahan et al., 2017) performs well if local training datasets are IID. Some methods (Karimireddy et al. (2020); Dieuleveut et al. (2021); Zhang et al. (2022); Guo et al. (2023b)) mitigate the negative impact of training data heterogeneity on convergence rate, while another branch ( Deng et al. (2020); Sharma et al. (2022); Sun & Wei (2022)) targets at reducing the performance bias of global model on local clients. Besides, few works ( Liu et al. (2021c); Nguyen et al. (2022); Guo et al. (2023c)) investigate the scenarios where the training data heterogeneity appears to be domain shift. All the above methods produce a shared global model which can diverge from the local optimal solutions when local target datasets are Non-IID.

**Personalized federated learning.** Many PFLs ( T Dinh et al. (2020); Hanzely et al. (2020); Fallah et al. (2020); Li et al. (2021); Tang et al. (2022); Cheng et al. (2023); Guo et al. (2023a)) train the personalized models with the guidance of a global model which embeds in the shared knowledge. Some researchers study the parameterized knowledge transfer between similar clients, e.g., MOCHA (Smith et al., 2017), FedAMP (Huang et al., 2021) and KT-pFL (Zhang et al., 2021). DFL (Luo et al., 2022) disentangles the shared features from the client-specific ones to achieve accurate aggregation on shared knowledge. Similarly, pFedPara (Hyeon-Woo et al., 2022) and Factorized-FL (Jeong & Hwang, 2022) factorizes the model parameters into the shared and personalized parts. Another branch ( Collins et al. (2021); Chen & Chao (2022); Xu et al. (2023)) employs the shared/aligned feature extractor to capture global knowledge and personalized classifiers to encode the personalization information. All of them don't cover the situations where there exists shortcut in local training datasets.

**Shortcut and Invariant learning (IL)** Causally invariant predictor is proposed in (Peters et al., 2016), and then applied into deep learning in IRM (Arjovsky et al., 2019) to mitigate shortcut. Subsequently, Rosenfeld et al. (2021) prove that IRM and its variants can be still trapped by shortcut when training environments are insufficient. IFM (Chen et al., 2022b) lowers the requirement and demands only logarithmic training environments. Some works focus on settling IL problem when the environment label is unavailable, e.g., EIIL (Creager et al., 2021), HRM (Liu et al., 2021a;b), EDNIL (Huang et al., 2022) and ZIN (Lin et al., 2022). Another branch (Ahuja et al. (2021); Chen et al. (2022a); Huh & Baidya (2022)) completes the constraints that IRM misses to improve the

performance. The iCaRL (Lu et al., 2022) extends IL to non-linear causal representations while ACTIR (Jiang & Veitch, 2022) extends IL to anti-causal scenarios. All these methods are devised for centralized scenarios where all training data is accessed and training environments are sufficient.

# B    COMPLETE THEORETICAL PROOFS

In this section, we will provide the complete proofs of the theorems stated in the main text.

## B.1    PROOF OF LEMMA 1

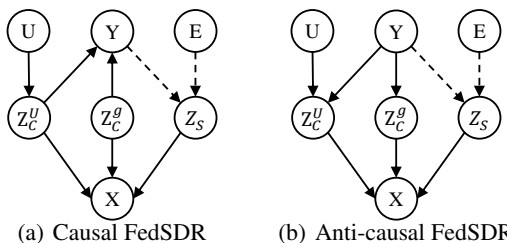

(a) Causal FedSDR          (b) Anti-causal FedSDR

Figure 4: The structural causal models (SCMs) considered in this paper.

From the above structural causal models, we derive the following two useful causal properties.

**Lemma 1**  *If the data generating mechanism of each federated client obeys the causal graph in Figure 4(a) or the anti-causal graph in Figure 4(b), we can have:*

- *$Z_S \perp\!\!\!\perp U \mid Y, E$ which means that the shortcut features $Z_S$ are conditionally independent of the personalization indicator $U$ given $Y$ and $E$.*

- *$Z_C^g \perp\!\!\!\perp Z_S \mid Y$ and $Z_C^U \perp\!\!\!\perp Z_S \mid Y$, which means that both the global ($Z_C^g$) and personalized ($Z_C^U$) invariant features are conditionally independent of the shortcut features $Z_S$ given $Y$.*

***Proof.***    According to the causal Markov condition (Theorem 1.4.1) proved in (Pearl, 2009), we know that the variable $Z_S$ is independent of all its nondescendants, given its parents in the (Markov) causal graph. Since $Y$ and $E$ are the parent variables of $Z_S$ and $U$ is a nondescendant of $Z_S$, the first causal signature in Lemma 1 is guaranteed. Besides, based on the $d$-separation criterion in (Pearl, 2009) we can find the variable $Y$ $d$-separates $Z_C^g$ from $Z_S$ and $d$-separates $Z_C^U$ from $Z_S$ in the SCMs. Therefore, we get the second causal signature in Lemma 1.

## B.2    PROOF OF THEOREM 1

**Theorem 1**  *If the Assumption 1 holds, the function $g$ is linear and the total number of training environments in the federated learning system satisfies $|\mathcal{E}_{tr}| > d_s$, then the following two statements are equivalent:*

- *$\Psi^\star(X)$ depends and only depends on the complete shortcut features $Z_S$;*

- *$\Psi^\star$ is the optima of the Eq.5 with an appropriately chosen hyper-parameter $\lambda$.*

***Proof.***    We write the linear feature extractors $\Psi$ that can recover the latent features ($[Z_C, Z_S]$) from the observation $X$ as $\Psi(X) = \Psi(g(Z_C, Z_S)) = AZ_C + BZ_S$, where $A$ and $B$ are fixed transformation matrices. This formulation is also adopted in the theoretical analysis in (Rosenfeld et al., 2021) and (Wang et al., 2022). For the concerned logistic regression, we can get a closed form

for the distribution $\mathbb{P}(Y \mid \Psi, e)$ as:

$$\mathbb{P}(Y \mid \Psi, e) := \mathbb{P}^e(Y \mid AZ_C + BZ_S)$$

$$= \frac{\mathbb{P}^e(AZ_C + BZ_S \mid Y)\mathbb{P}^e(Y)}{\mathbb{P}^e(AZ_C + BZ_S)}$$

$$= \frac{\mathbb{P}^e(AZ_C + BZ_S \mid Y)\mathbb{P}^e(Y)}{\sum_y \mathbb{P}^e(Y = y)\mathbb{P}^e(AZ_C + BZ_S \mid Y = y)}$$

Since Assumption 1 holds, we have $\mathbb{P}^e(Y = y) = \mathbb{P}^e(Y = y'), \forall y \in Y$. We can derive that

$$\mathbb{P}^e(y \mid AZ_C + BZ_S) = \frac{\mathbb{P}^e(AZ_C + BZ_S \mid y)}{\sum_y \mathbb{P}^e(AZ_C + BZ_S \mid Y = y)}$$

$$= \frac{\mathbb{P}^e(AZ_C + BZ_S \mid y)}{\mathbb{P}^e(AZ_C + BZ_S \mid Y = y) + \mathbb{P}^e(AZ_C + BZ_S \mid Y = -y)}$$

$$= \frac{1}{1 + \frac{\mathbb{P}^e(AZ_C + BZ_S \mid Y = -y)}{\mathbb{P}^e(AZ_C + BZ_S \mid Y = y)}}, \quad \forall y \in \{\pm 1\}.$$

Because we have $Z_C \perp\!\!\!\perp Z_S \mid Y$ from Theorem 1, we can get the probability density of $AZ_C + BZ_S$ as follows:

$$AZ_C + BZ_S \mid y \sim \mathcal{N}(y \cdot \mu_z, \Sigma_z), \tag{8}$$

where $\mu_z = A\mu_c + B\mu_s$ and $\Sigma_z = AA^T\sigma_c^2 + BB^T\sigma_s^2$. Thus, we can get the $\mathbb{P}(Y \mid \Psi, e)$ as:

$$\mathbb{P}^e(y \mid \Psi) = \frac{1}{1 + \frac{\mathbb{P}^e(AZ_C + BZ_S \mid Y = -y)}{\mathbb{P}^e(AZ_C + BZ_S \mid Y = y)}}$$

$$= \frac{1}{1 + \exp(-y \cdot 2\Psi^T\Sigma_z^{-1}\mu_z)}, \quad \forall y \in \{\pm 1\},$$

where $\Sigma_z^{-1}$ represents the generalized inverse of $\Sigma_z$, i.e., $\Sigma_z^{-1}\Sigma_z = I$.

According to Lemma F.2. proved in the appendix of (Rosenfeld et al., 2021), the optimal classifier based on the feature extractor $\Psi(X) = AZ_C + BZ_S$ is sufficiently and necessarily given by $2(AA^T\sigma_c^2 + BB^T\sigma_s^2)^{-1}(A\mu_c + B\mu_s)$. That is, we have $\mathbb{P}_{\omega_i^\star}(y \mid \Psi) = \frac{1}{1 + \exp(-y \cdot 2\Psi^T\Sigma_z^{-1}\mu_z)}$, $\forall y \in \{\pm 1\}$, if and only if $\omega_i^\star \in \arg\min_{\omega_i: \mathcal{H} \to \mathcal{Y}} \sum_{u=1}^N \rho_u^i \mathcal{R}(\omega_i(\Psi); e_i), \forall e_i \in \mathcal{E}_{tr}$.

Therefore, we can calculate the KL-divergence between $\mathbb{P}_{\omega_i^\star}(Y \mid \Psi, e_i)$ and $\mathbb{P}_{\omega_j^\star}(Y \mid \Psi, e_j)$ by

$$\mathcal{KL}\big(\mathbb{P}_{\omega_i^\star}(Y \mid \Psi, e_i)\big\|\mathbb{P}_{\omega_j^\star}(Y \mid \Psi, e_j)\big)$$

$$= \sum_{y \in \{\pm 1\}} \mathbb{P}_{\omega_i^\star}(y \mid \Psi, e_i) \log \frac{\mathbb{P}_{\omega_i^\star}(y \mid \Psi, e_i)}{\mathbb{P}_{\omega_j^\star}(y \mid \Psi, e_j)}$$

$$= \sum_{y \in \{\pm 1\}} \frac{1}{1 + \exp(-y \cdot 2\Psi^T\Sigma_{z_i}^{-1}\mu_z^i)} \log \frac{1 + \exp(-y \cdot 2\Psi^T\Sigma_{z_j}^{-1}\mu_z^j)}{1 + \exp(-y \cdot 2\Psi^T\Sigma_{z_i}^{-1}\mu_z^i)}$$

$$= \frac{1}{1 + \exp(-2\Psi^T\Sigma_{z_i}^{-1}\mu_z^i)} \log \frac{1 + \exp(-2\Psi^T\Sigma_{z_j}^{-1}\mu_z^j)}{1 + \exp(-2\Psi^T\Sigma_{z_i}^{-1}\mu_z^i)}$$

$$+ \frac{1}{1 + \exp(2\Psi^T\Sigma_{z_i}^{-1}\mu_z^i)} \log \frac{1 + \exp(2\Psi^T\Sigma_{z_j}^{-1}\mu_z^j)}{1 + \exp(2\Psi^T\Sigma_{z_i}^{-1}\mu_z^i)}$$

$$= \frac{1}{1 + \exp(-2\Psi^T\Sigma_{z_i}^{-1}\mu_z^i)} \left\{ \log \frac{1 + \exp(2\Psi^T\Sigma_{z_j}^{-1}\mu_z^j)}{1 + \exp(2\Psi^T\Sigma_{z_i}^{-1}\mu_z^i)} + \log \frac{\exp(2\Psi^T\Sigma_{z_i}^{-1}\mu_z^i)}{\exp(2\Psi^T\Sigma_{z_j}^{-1}\mu_z^j)} \right\}$$

$$+ \frac{1}{1 + \exp(2\Psi^T\Sigma_{z_i}^{-1}\mu_z^i)} \log \frac{1 + \exp(2\Psi^T\Sigma_{z_j}^{-1}\mu_z^j)}{1 + \exp(2\Psi^T\Sigma_{z_i}^{-1}\mu_z^i)}$$

$$= \log \frac{1 + \exp(2\Psi^T\Sigma_{z_j}^{-1}\mu_z^j)}{1 + \exp(2\Psi^T\Sigma_{z_i}^{-1}\mu_z^i)} + \frac{2\Psi^T(\Sigma_{z_i}^{-1}\mu_z^i - \Sigma_{z_j}^{-1}\mu_z^j)}{1 + \exp(-2\Psi^T\Sigma_{z_i}^{-1}\mu_z^i)}$$

Similarly, we can get that

$$\mathcal{KL}\big(\mathbb{P}_{\omega_j^\star}(Y \mid \Psi, e_j) \big\| \mathbb{P}_{\omega_i^\star}(Y \mid \Psi, e_i)\big)$$

$$= \sum_{y \in \{\pm 1\}} \mathbb{P}_{\omega_j^\star}(y \mid \Psi, e_j) \log \frac{\mathbb{P}_{\omega_j^\star}(y \mid \Psi, e_j)}{\mathbb{P}_{\omega_i^\star}(y \mid \Psi, e_i)}$$

$$= \log \frac{1 + \exp(2\Psi^T \Sigma_{z_i}^{-1} \mu_z^i)}{1 + \exp(2\Psi^T \Sigma_{z_j}^{-1} \mu_z^j)} + \frac{2\Psi^T (\Sigma_{z_j}^{-1} \mu_z^j - \Sigma_{z_i}^{-1} \mu_z^i)}{1 + \exp(-2\Psi^T \Sigma_{z_j}^{-1} \mu_z^j)}$$

Combining the above results, we can get that

$$\mathcal{KL}\big(\mathbb{P}_{\omega_i^\star}(Y \mid \Psi, e_i) \big\| \mathbb{P}_{\omega_j^\star}(Y \mid \Psi, e_j)\big) + \mathcal{KL}\big(\mathbb{P}_{\omega_j^\star}(Y \mid \Psi, e_j) \big\| \mathbb{P}_{\omega_i^\star}(Y \mid \Psi, e_i)\big)$$

$$= \underbrace{\left\{ \frac{1}{1 + \exp(-2\Psi^T \Sigma_{z_i}^{-1} \mu_z^i)} - \frac{1}{1 + \exp(-2\Psi^T \Sigma_{z_j}^{-1} \mu_z^j)} \right\}}_{T_1} \cdot \underbrace{\left\{ 2\Psi^T (\Sigma_{z_i}^{-1} \mu_z^i - \Sigma_{z_j}^{-1} \mu_z^j) \right\}}_{T_2}$$

$$\geq 0, \forall e_i, e_j \in \mathcal{E}_{all}, \forall \Psi \in \mathcal{H}.$$

Since the absolute value of term $T_1$ (i.e., $|T_1|$) monotonically increases with term $|T_2|$ increasing, the objective $\max_\Psi \mathcal{KL}\big(\mathbb{P}_{\omega_i^\star}(Y \mid \Psi, e_i) \big\| \mathbb{P}_{\omega_j^\star}(Y \mid \Psi, e_j)\big) + \mathcal{KL}\big(\mathbb{P}_{\omega_j^\star}(Y \mid \Psi, e_j) \big\| \mathbb{P}_{\omega_i^\star}(Y \mid \Psi, e_i)\big)$ is equivalent to $\max_\Psi \big\| 2\Psi^T (\Sigma_{z_i}^{-1} \mu_z^i - \Sigma_{z_j}^{-1} \mu_z^j) \big\|^2$. Therefore, the second term $\frac{1}{N} \sum_{u=1}^N \ell_{dis}(\Psi; D_u)$ in Eq.5 can be written as

$$\frac{1}{N} \sum_{u=1}^N \ell_{dis}(\Psi; D_u) = \mathbb{E}_\Psi \Big[ \sum_{e_i \in \mathcal{E}_{tr}} \sum_{e_j \in \mathcal{E}_{tr}} \big\| 2\Psi^T (\Sigma_{z_i}^{-1} \mu_z^i - \Sigma_{z_j}^{-1} \mu_z^j) \big\|^2 \Big]$$

$$= \sum_{e_i \in \mathcal{E}_{tr}} \sum_{e_j \in \mathcal{E}_{tr}} \frac{4\|(A\mu_c^i + B\mu_s^i) - (A\mu_c^j + B\mu_s^j)\|^2}{(AA^T \sigma_c^2 + BB^T \sigma_s^2)^2} \cdot \mathbb{E}_\Psi \|\Psi\|^2$$

$$= \sum_{e_i \in \mathcal{E}_{tr}} \sum_{e_j \in \mathcal{E}_{tr}} \frac{4\|B(\mu_s^i - \mu_s^j)\|^2}{(AA^T \sigma_c^2 + BB^T \sigma_s^2)^2} \cdot \mathbb{E}_\Psi \|\Psi\|^2$$

According to the mentioned $AZ_C + BZ_S \mid y \sim \mathcal{N}(y \cdot \mu_z, \Sigma_z)$, we can get the density

$$\mathbb{P}(\Psi) = \sum_{y \in Y} \mathbb{P}(Y = y) \mathbb{P}(\Psi \mid Y = y)$$

With the Assumption 1 holding, we can get the mean $\mathbb{E}[\Psi] = 0$ and the variance $\mathbb{D}[\Psi] = AA^T \sigma_c^2 + BB^T \sigma_s^2$. Therefore, we have

$$\frac{1}{N} \sum_{u=1}^N \ell_{dis}(\Psi; D_u) = \mathbb{E}_\Psi \Big[ \sum_{e_i \in \mathcal{E}_{tr}} \sum_{e_j \in \mathcal{E}_{tr}} \big\| 2\Psi^T (\Sigma_{z_i}^{-1} \mu_z^i - \Sigma_{z_j}^{-1} \mu_z^j) \big\|^2 \Big]$$

$$= \sum_{e_i \in \mathcal{E}_{tr}} \sum_{e_j \in \mathcal{E}_{tr}} \frac{4\|B(\mu_s^i - \mu_s^j)\|^2}{(AA^T \sigma_c^2 + BB^T \sigma_s^2)^2} \cdot \{\mathbb{D}[\Psi] + (\mathbb{E}[\Psi])^2\}$$

$$= \sum_{e_i \in \mathcal{E}_{tr}} \sum_{e_j \in \mathcal{E}_{tr}} \frac{4\|B(\mu_s^i - \mu_s^j)\|^2}{AA^T \sigma_c^2 + BB^T \sigma_s^2}$$

$$= \frac{4}{\frac{AA^T}{BB^T} \sigma_c^2 + \sigma_s^2} \sum_{e_i \in \mathcal{E}_{tr}} \sum_{e_j \in \mathcal{E}_{tr}} \|\mu_s^i - \mu_s^j\|^2$$

We can find that maximizing the above objective will make $A = 0$ and $BB^T \neq 0$. Moreover, when $|\mathcal{E}_{tr}| > d_s$, maximizing $\sum_{e_i \in \mathcal{E}_{tr}} \sum_{e_j \in \mathcal{E}_{tr}} \|\mu_s^i - \mu_s^j\|^2$ will make $rank(B) = d_s$. In the meanwhile, satisfying $A = 0$ and $rank(B) = d_s$ will in turn maximize the objective $\frac{1}{N} \sum_{u=1}^N \ell_{dis}(\Psi; D_u)$.

In Eq.5, we utilize a Lagrangian multiplier to solve the constrained optimization and the balancing weight is $\lambda$. Therefore, Theorem 2 gets proved.

### B.3 PROOF OF THEOREM 2

**Theorem 2** *Suppose $\Psi^\star(X)$ in the Eq.7 depends and only depends on the complete shortcut features $Z_S$. If $f_{\theta_u}^\star$ ($\forall u \in [N]$) is the optima of the Eq.7 with the hyper-parameter $\gamma$ chosen appropriately, then the $f_{\theta_u}^\star$ is the optimal personalized invariant predictor for the client $u$, $\forall u \in [N]$.*

**Proof.** We know that minimizing $\mathcal{R}(\omega_u(\Phi_u); D_u)$ is the sufficient condition of maximizing $I(Y; \Phi_u(X))$, and $I(\Phi_u; \Psi^\star \mid Y) = 0$ is equivalent to $\Phi_u \perp\!\!\!\perp \Psi^\star \mid Y$. According to the property of Lagrangian multiplier, the objective in Eq.7 is equivalent to the constrained optimization where the constrain is $I(\Phi_u; \Psi^\star \mid Y) = 0$, with the appropriately chosen $\gamma$. Combining with the second causal signature in Theorem 1, Theorem 3 gets proved.

## C MORE DETAILS OF EXPERIMENTS

In this section, we will include more detailed setups and discussions on the evaluation part. Code is available at https://github.com/Tangx-yy/FedSDR.

### C.1 NON-IID DATA PARTITION

For CMNIST and CFMNIST datasets, we provide two training environments ($p_{train}^e = 0.90$ and $0.80$) as $\mathcal{E}_{train}$ and every local client only has one training environment which is randomly sampled from the training environment set $\mathcal{E}_{train}$. To assess the model performance on different test distributions, the test environment on each client varies across $p_{test}^e = 0.00, 0.10, ..., 0.90, 1.00$. Considering the heterogeneous data generating process across local clients, the data instances used for constructing the training/test environments on each client are randomly sampled from only two digit sub-classes (1 separated and 1 overlapped) labeled 0 and two digit sub-classes (1 separated and 1 overlapped) labeled 1 without replacement. Specifically, we totally simulate eight local clients and one server in the federated learning system. For example, the data instances on client 1 are randomly sampled from digit 0, 1, 5, 6; the data instances on client 2 are randomly sampled from digit 1, 2, 6, 7; the data instances on client 3 are randomly sampled from digit 2, 3, 7, 8; and the data instances on client 8 are randomly sampled from digit 3, 4, 8, 9.

As regard to WaterBird, we distribute 15 (10 separated and 5 overlapped) waterbird species and 51 (34 separated and 17 overlapped) landbird species to each local client. Both the training and test data instances are constructed using bird photographs randomly sampled from the corresponding bird species in the bird dataset and background photographs randomly selected from the background dataset without replacement. Similarly, we totally simulate eight local clients and one server in the federated learning system.

PACS consists of 7 classes (i.e., dog, elephant, giraffe, guitar, horse, house, and person) distributed across 4 domains/environments (i.e., Art Painting, Cartoon, Photo and Sketch). We adopt the "leave-one-domain-out" strategy to evaluate the out-of-distribution (OOD) generalization performance. For example, when we evaluate the performance on Art Painting domain, we use the remaining three domains (i.e., Cartoon, Photo and Sketch) as training environments. Taking personalization into consideration, we split each training domain into two subsets according to classes (i.e., one subset consists of dog, elephant and giraffe and another subset consists of guitar, horse, house, and person), and then distribute these two subsets onto two clients respectively. The training and test datasets on each client come from different domains but consist of the same classes.

### C.2 HYPER-PARAMETERS

The hyper-parameters of the competitors and our algorithm are tuned to make the accuracy on the validation environment (i.e., $p_{val}^e = 0.10$) as high as possible. Specifically, the mainly used hyper-parameters in the evaluation part are listed as follows: Global communication round: $T = 600$, Local iterations: $R = 10$, Personalized epochs to update the personalized invariant predictors: $K = 10$, Local batch size: $B = 50$, Global learning rate: $\beta = 0.0001$, Personalized learning rate: $\eta = 0.0001$, Discrepancy threshold: $\alpha = 1.0$, Balancing weight: $\lambda = 0.5$, Balancing weight: $\gamma = 1.4$, Optimizer: Adam.

### C.3 IMPLEMENTATION

Besides, the experiments are implemented in PyTorch. We simulate a set of clients and a centralized server on one deep learning workstation (Intel(R) Core(TM) i9-12900K CPU @ 3.20GHz with one NVIDIA GeForce RTX 3090 GPU).

### C.4 ALGORITHMS

---

**Algorithm 1** FedSDR: Federated Learning with Shortcut Discovery and Removal

---

**Input**: $T, R, K, \beta, \eta, \alpha, \lambda, \gamma$.

1: Initialize the models $\Psi^0$, $\{\omega_i^0 | i \in [|\mathcal{E}_{tr}|]\}$, $\{f_{\theta_u}^0 | u \in [N]\}$.
2: **for** $t = 0$ to $T - 1$ **do**
3:     Server sends the global models $(\Psi^t, \{\omega_i^t | i \in [|\mathcal{E}_{tr}|]\})$ to the participating local clients.
4:     **for** local device $u = 1$ to $N$ in parallel **do**
5:         Initialization: $\Psi_u^{t,0} \leftarrow \Psi^t$, $\{\omega_{i,u}^t \leftarrow \omega_i^t | i \in [|\mathcal{E}_{tr}^u|]\}$.
6:         **for** $k = 0$ to $K - 1$ **do**
7:             Update the personalized invariant model: $f_{\theta_u}^{t,k+1} = f_{\theta_u}^{t,k} - \eta \nabla \ell_{SR}^u(f_{\theta_u}^{t,k}; D_u)$.
8:         $f_{\theta_u}^{t+1,0} \leftarrow f_{\theta_u}^{t,K}$.
9:         **for** $r = 0$ to $R - 1$ **do**
10:            Update the shortcut extractor: $\Psi_u^{t,r+1} = \Psi_u^{t,r} - \beta \nabla \ell_{SD}^u(\Psi_u^{t,r}; D_u)$.
11:            Update the environment classifiers for $K$ epochs with $\nabla \mathcal{R}(\omega_{i,u}^{t,r}(\Psi_u^{t,r}); e_i)$.
12:     Randomly select a subset $(S^t)$ of the users to upload the local approximation:
13:     $\Psi_u^{t+1} \leftarrow \Psi_u^{t,R}$ and $\{\omega_{i,u}^{t+1} \leftarrow \omega_{i,u}^{t,R} \mid i = 1, 2, ..., |\mathcal{E}_{tr}^u|\}$.
14:     Global aggregation: shortcut extractor $\Psi^{t+1} = \frac{1}{|S^t|} \sum_{u \in S^t} \Psi_u^{t+1}$ and environment classifiers $\omega_i^{t+1} = \sum_{u \in S^t} \frac{\rho_u^i}{\sum_{u \in S^t} \rho_u^i} \omega_{i,u}^{t+1}, i = 1, 2, ..., |\mathcal{E}_{tr}|$.
15: **return** the personalized invariant models $\{f_{\theta_u}^{T,0} | u \in [N]\}$.

---

In the evaluation part, the data distributions among the local clients only overlap slightly. Therefore, the personalized invariant models are trained and updated locally and do not participate in the global aggregation. The corresponding algorithm is shown in Algorithm 1.

---

**Algorithm 2** FedSDR (+FedAvg): Federated Learning with Shortcut Discovery and Removal

---

**Input**: $T, R, K, \beta, \eta, \alpha, \lambda, \gamma$.

1: Initialize the models $\Psi^0$, $f_\theta^0$ and $\{\omega_i^0 | i \in [|\mathcal{E}_{tr}|]\}$.
2: **for** $t = 0$ to $T - 1$ **do**
3:     Server sends the global models $(\Psi^t, f_\theta^t$ and $\{\omega_i^t | i \in [|\mathcal{E}_{tr}|]\})$ to the participating local clients.
4:     **for** local device $u = 1$ to $N$ in parallel **do**
5:         Initialization: $\Psi_u^{t,0} \leftarrow \Psi^t$, $f_{\theta_u}^{t,0} \leftarrow f_\theta^t$ and $\{\omega_{i,u}^t \leftarrow \omega_i^t | i \in |\mathcal{E}_{tr}^u|\}$.
6:         **for** $r = 0$ to $R - 1$ **do**
7:            Update the personalized invariant model: $f_{\theta_u}^{t,r+1} = f_{\theta_u}^{t,r} - \eta \nabla \ell_{SR}^u(f_{\theta_u}^{t,r}; D_u)$.
8:            Update the shortcut extractor: $\Psi_u^{t,r+1} = \Psi_u^{t,r} - \beta \nabla \ell_{SD}^u(\Psi_u^{t,r}; D_u)$.
9:            Update the environment classifiers for $K$ epochs with $\nabla \mathcal{R}(\omega_{i,u}^{t,r}(\Psi_u^{t,r}); e_i)$.
10:     Randomly select a subset $(S^t)$ of the users to upload the local approximation:
11:     $\Psi_u^{t+1} \leftarrow \Psi_u^{t,R}$, $f_{\theta_u}^{t+1} \leftarrow f_{\theta_u}^{t,R}$ and $\{\omega_{i,u}^{t+1} \leftarrow \omega_{i,u}^{t,R} \mid i = 1, 2, ..., |\mathcal{E}_{tr}^u|\}$.
12:     Global aggregation: the shortcut extractor $\Psi^{t+1} = \frac{1}{|S^t|} \sum_{u \in S^t} \Psi_u^{t+1}$, the environment classifiers $\omega_i^{t+1} = \sum_{u \in S^t} \frac{\rho_u^i}{\sum_{u \in S^t} \rho_u^i} \omega_{i,u}^{t+1}, i = 1, 2, ..., |\mathcal{E}_{tr}|$, and the global invariant model $f_\theta^{t+1} = \frac{1}{|S^t|} \sum_{u \in S^t} f_{\theta_u}^{t+1}$.
13: **return** the global and personalized invariant models $f_\theta^T$, $\{f_{\theta_u}^{T,R} | u \in [N]\}$.

---

As mentioned in the main text, the proposed shortcut discovery and removal method can easily cooperate with most of the existing federated and personalized federated learning method to improve

the out-of-distribution generalization performance via adding shortcut discovery and removal as a regularization term. We provide an example combination of FedSDR with FedAvg (McMahan et al., 2017) and pFedMe (T Dinh et al., 2020) in Algorithm 2 and Algorithm 3, respectively. Of course, the combinations with more other federated and personalized federated learning methods can be explored in the future.

---

**Algorithm 3** FedSDR (+pFedMe): Federated Learning with Shortcut Discovery and Removal

---

**Input**: $T, R, K, \beta, \eta, \alpha, \lambda, \gamma$.

1: Initialize the models $\Psi^0$, $f_\theta^0$ and $\{\omega_i^0 | i \in [|\mathcal{E}_{tr}|]\}$.
2: **for** $t = 0$ to $T - 1$ **do**
3:   Server sends the global models ($\Psi^t$, $f_\theta^t$ and $\{\omega_i^t | i \in [|\mathcal{E}_{tr}|]\}$) to the participating local clients.
4:   **for** local device $u = 1$ to $N$ in parallel **do**
5:     Initialization: $\Psi_u^{t,0} \leftarrow \Psi^t$, $f_\theta^{t,0} \leftarrow f_\theta^t$ and $\{\omega_{i,u}^t \leftarrow \omega_i^t | i \in |\mathcal{E}_{tr}^u|\}$.
6:     **for** $r = 0$ to $R - 1$ **do**
7:       **for** $k = 0$ to $K - 1$ **do**
8:         Update the personalized invariant model:
9:           $f_{\theta_u}^{r,k+1} = f_{\theta_u}^{r,k} - \eta(\nabla \ell_{SR}^u(f_{\theta_u}^{r,k}; D_u) + \gamma(f_{\theta_u}^{r,k} - f_\theta^{t,r}))$.
10:       Update the global invariant model: $f_\theta^{t,r+1} = f_\theta^{t,r} - \beta\gamma(f_\theta^{t,r} - f_{\theta_u}^{r,K})$
11:       Update the shortcut extractor: $\Psi_u^{t,r+1} = \Psi_u^{t,r} - \beta\nabla\ell_{SD}^u(\Psi_u^{t,r}; D_u)$.
12:       Update the environment classifiers for $K$ epochs with $\nabla\mathcal{R}(\omega_{i,u}^{t,r}(\Psi_u^{t,r}); e_i)$.
13:     Randomly select a subset ($S^t$) of the users to upload the local approximation:
14:     $\Psi_u^{t+1} \leftarrow \Psi_u^{t,R}$, $f_{\theta_u}^{t+1} \leftarrow f_{\theta_u}^{t,R}$ and $\{\omega_{i,u}^{t+1} \leftarrow \omega_{i,u}^{t,R} \mid i = 1, 2, ..., |\mathcal{E}_{tr}^u|\}$.
15:     Global aggregation: the shortcut extractor $\Psi^{t+1} = \frac{1}{|S^t|}\sum_{u \in S^t} \Psi_u^{t+1}$, the environment classifiers $\omega_i^{t+1} = \sum_{u \in S^t} \frac{\rho_u^i}{\sum_{u \in S^t} \rho_u^i}\omega_{i,u}^{t+1}, i = 1, 2, ..., |\mathcal{E}_{tr}|$, and the global invariant model $f_\theta^{t+1} = \frac{1}{|S^t|}\sum_{u \in S^t} f_{\theta_u}^{t+1}$.
16: **return** the global and personalized invariant models $f_\theta^T$, $\{f_{\theta_u}^{R,K} | u \in [N]\}$.

---

## C.5 MORE EXPERIMENTAL RESULTS

**Experiment on synthetic dataset**    To verify the provided theoretical guarantees under linear cases, we generate a synthetic dataset using the same strategy as in (Rosenfeld et al., 2021). Specifically, it is a logistic regression task and the data instance $X$ is generated by $X = g(Z_C^g, Z_C^U, Z_S)$, where the dimensionalities of $Z_C^g$, $Z_C^U$ and $Z_S$ are $d_C^g = 3$, $d_C^U = 3$ and $d_S = 6$ respectively. The linear function $g$ is implemented by one fully-connected layer which has 12 neurons. The latent variables $Z_C^g$, $Z_C^U$ and $Z_S$ are subject to $\mathcal{N}(y \cdot \mu_{c,g}, \sigma_{c,g}^2 I)$, $\mathcal{N}(y \cdot \mu_{c,u}, \sigma_{c,u}^2 I)$ and $\mathcal{N}(y \cdot \mu_s, \sigma_s^2 I)$ respectively. Target variable $y$ is taken from the distribution $\mathbb{P}(y = -1) = \mathbb{P}(y = 1) = 0.5$. Both $\mu_{c,g}$ and $\mu_{c,u}$ are randomly sampled from $\mathcal{N}(0, 1.5I)$ while $\mu_s$ is randomly sampled from $\mathcal{N}(0, 0.75I$. To make the shortcut representation $Z_S$ easier to learn, we choose $\sigma_{c,g} = \sigma_{c,u} = 2$ and $\sigma_s = 1$ as in (Rosenfeld et al., 2021). Each fixed value of $\mu_s$ indicates one specified environment. We generate 10 training environments and 5000 test environments to evaluate the out-of-distribution generalization performance. Each (training/test) environment contains 10000 data samples $(X, y)$ and the training data samples are distributed onto totally 100 clients. The training and test data samples on each client are generated with an identical value of $\mu_{c,u}$. Besides, we choose the client sampling rate as 0.1. The experimental results on this synthetic dataset are shown in Table 4: In

Table 4: The performance of FedSDR and the competitors on the synthetic dataset.

| Algorithm | FedAvg | DRFA | FedSR | FedIIR | FTFA | pFedMe | Ditto | FedRep | FedRoD | FedPAC | **FedSDR** |
|---|---|---|---|---|---|---|---|---|---|---|---|
| worst-case(%) | 3.06 | 62.41 | 63.09 | 67.39 | 1.32 | 10.58 | 7.76 | 2.57 | 21.53 | 8.98 | 92.49 |
| average(%) | 85.56 | 69.64 | 70.53 | 70.75 | 96.26 | 95.72 | 96.50 | 97.80 | 97.24 | 98.77 | 96.07 |

particular, when we manually select the causal features $[Z_C^g, Z_C^U]$ as the discriminating features, we find the optimal personalized classifiers achieve an stable accuracy around 97.5 in different test

environments. Therefore, the results shown in Table 4 can demonstrate the effectiveness of our FedSDR on developing the optimal personalized invariant predictors, compared with the state-of-the-art FL and PFL methods.

**Scalability of FedSDR**  In the evaluation part of the main text, we simulate 8 clients in the experiments on CMNIST, CFMNIST and WaterBird. The experiments on PACS are conducted on 6 clients. To further evaluate the scalability of FedSDR, we firstly partition CMNIST dateset into 8 subsets using the same strategy adopted to simulate 8 clients. And then, we randomly distribute each subset onto 10 clients. In this way, we totally construct 80 clients for CMNIST dataset. Similarly, we construct 80, 80, 60 clients for CFMNIST, WaterBird and PACS respectively. When evaluating the model performance on these four datasets, we adopt a client sampling rate of 0.1. The experimental results are shown in Table:

Table 5: The overall comparison between the performance of our method FedSDR and the baselines with a large number of clients.

| Dataset | CMNIST | | CFMNIST | | WaterBird | | PACS | |
|---|---|---|---|---|---|---|---|---|
| Test accuracy (%) | worst-case | average | worst-case | average | worst-case | average | worst-case | average |
| FedAvg | 1.74 | 46.82 | 0.77 | 45.62 | 48.65 | 61.57 | 33.75 | 40.18 |
| DRFA | 14.94 | 47.24 | 15.51 | 47.14 | 52.34 | 60.43 | 36.17 | 41.75 |
| FedSR | 40.29 | 43.64 | 41.16 | 43.27 | 55.63 | 64.32 | 39.03 | 43.40 |
| FedIIR | 41.18 | 42.93 | 41.80 | 43.58 | 54.31 | 64.60 | 40.15 | 44.37 |
| FTFA | 11.51 | 49.28 | 7.20 | 47.57 | 50.25 | 63.39 | 34.65 | 42.19 |
| pFedMe | 17.28 | 44.13 | 2.42 | 47.95 | 50.01 | 61.97 | 41.06 | 45.84 |
| Ditto | 1.98 | 45.84 | 1.80 | 45.71 | 49.08 | 63.38 | 40.18 | 46.30 |
| FedRep | 1.56 | 46.20 | 0.83 | 46.14 | 48.12 | 64.52 | 42.16 | 47.58 |
| FedRoD | 6.53 | 46.86 | 1.60 | 47.43 | 49.56 | 65.49 | 42.68 | 46.61 |
| FedPAC | 0.38 | 45.64 | 0.23 | 44.88 | 42.61 | 63.81 | 44.19 | 49.71 |
| **FedSDR** | **50.41** | **51.85** | **52.81** | **57.14** | **59.96** | **68.09** | **48.07** | **51.55** |

The results show that FedSDR can still outperform the competitors when there are a large number of clients in the federated learning system, which can validate the scalability of the proposed FedSDR.

**More invariant learning methods**  We consider combinations of the personalized federated learning schemes with more state-of-the-art invariant learning methods, including IRM-IB (Ahuja et al., 2021) and MRI (Huh & Baidya, 2022). The experimental results are shown in Table 6. We can find that our algorithm FedSDR can outperform these combinations on both worst-case and average-case performances.

Table 6: The comparison between the performance of FedSDR and the centralized invariant learning.

| Dataset | CMNIST | | CFMNIST | | WaterBird | |
|---|---|---|---|---|---|---|
| Test case | worst-case (%) | average (%) | worst-case (%) | average (%) | worst-case (%) | average (%) |
| IRM[†] | 46.38(±0.61) | 49.14(±0.89) | 47.76(±0.68) | 49.41(±81) | 60.38(±1.54) | 68.63(±1.91) |
| IRM-IB[†] | 48.15(±0.34) | 49.78(±0.72) | 49.04(±0.52) | 50.54(±66) | 60.75(±1.43) | 69.82(±1.64) |
| MRI[†] | 46.81(±0.49) | 48.92(±0.84) | 48.84(±0.61) | 49.33(±93) | 61.64(±1.02) | 69.47(±1.35) |
| IRM[†]-FT | 14.32(±0.44) | 54.27(±0.74) | 11.09(±0.81) | 53.48(±0.74) | 60.25(±0.78) | 69.46(±1.25) |
| IRM[†]-L2 | 45.68(±0.58) | 49.04(±0.83) | 47.92(±0.42) | 49.46(±0.92) | 61.25(±0.86) | 68.93(±1.46) |
| IRM-IB[†]-FT | 17.21(±0.23) | 54.35(±0.56) | 13.47(±0.38) | 53.14(±0.52) | 60.84(±0.64) | 70.31(±1.06) |
| IRM-IB[†]-L2 | 47.79(±0.45) | 50.13(±0.67) | 49.73(±0.51) | 51.65(±0.75) | 61.38(±0.86) | 69.08(±1.13) |
| MRI[†]-FT | 15.52(±0.61) | 53.98(±0.92) | 12.13(±0.79) | 54.21(±0.88) | 61.14(±0.93) | 70.84(±1.64) |
| MRI[†]-L2 | 47.21(±0.73) | 52.37(±0.87) | 48.73(±0.85) | 50.67(±0.87) | 62.18(±0.96) | 69.16(±1.71) |
| **FedSDR** | **53.88** (±0.24) | **55.59** (±0.65) | **56.92** (±0.73) | **61.88** (±1.02) | **65.25** (±0.97) | **73.20** (±1.28) |

**Compatibility of FedSDR with typical PFL schemes**  We implement Algorithm 2 and Algorithm 3 mentioned above on three datasets (i.e., Colored-MNIST, Colored-FMNIST and WaterBird). The values of the generic hyper- parameters are set same as FedSDR. In particular, we choose $\lambda = 0.5$ and $\gamma = 1.0 * 10^4$ for Colored-MNIST and Colored-FMNIST. As to WaterBird dataset, we choose $\lambda = 0.5$ and $\gamma = 1.4$. The experimental results are shown in the following Table 7. Note that the performances of FedSDR+FedAvg and FedSDR+pFedMe are evaluated with the personalized invariant models output by Algorithm 2 and Algorithm 3, respectively. The results show that the model performance can be further improved when we combine the proposed shortcut discovery and removal method with the prevalent federated learning algorithms.

Table 7: The combinations of our method with other federated learning schemes.

| Dataset | CMNIST | | CFMNIST | | WaterBird | |
|---|---|---|---|---|---|---|
| Test case | worst-case (%) | average (%) | worst-case (%) | average (%) | worst-case (%) | average (%) |
| FedSDR | 53.88($\pm$0.24) | 55.59($\pm$0.65) | 56.92($\pm$0.73) | 61.88($\pm$1.02) | 65.25($\pm$0.97) | 73.20($\pm$1.28) |
| FedSDR+FedAvg | 51.76($\pm$0.28) | 56.01($\pm$0.33) | 57.14($\pm$0.85) | 61.56($\pm$0.92) | 66.73($\pm$1.13) | 74.27($\pm$1.45) |
| FedSDR+pFedMe | 53.54($\pm$0.19) | 55.93($\pm$0.26) | 56.69($\pm$0.68) | 63.22($\pm$0.76) | 67.32($\pm$0.81) | 74.38($\pm$1.05) |

**The effect of hyper-parameters**  As discussed in the main text, there are two significant hyper-parameters in the proposed objective 5 and 6, i.e., $\lambda$ and $\gamma$. We will evaluate the effects of these two hyper-parameters on the performance of our method FedSDR. When we evaluate the effect of the hyper-parameter $\lambda$, we fix $\gamma = 1.0 * 10^4$ for CMNIST and CFMNIST datasets, and $\gamma = 1.4$ for WaterBird dataset. When we evaluate the effect of the hyper-parameter $\gamma$, we fix $\lambda = 0.5$ for CMNIST and CFMNIST datasets, and $\lambda = 0.5$ for WaterBird dataset. The results on CMNIST. CFMNIST and WaterBird are given in Table 8, Table 9 and Table 10, respectively.

Table 8: The performance of FedSDR on CMNIST dataset with different hyper-parameters $\lambda$ and $\gamma$.

| $\lambda$ | 0.10 | 0.20 | 0.30 | 0.40 | 0.50 | 0.60 | 0.70 | 0.80 | 0.90 | 1.00 | 10.0 |
|---|---|---|---|---|---|---|---|---|---|---|---|
| worst-case (%) | 53.73 | 53.62 | 53.18 | 53.93 | 53.88 | 53.51 | 53.83 | 53.66 | 53.79 | 53.82 | 52.35 |
| average (%) | 55.08 | 55.37 | 55.72 | 55.68 | 55.59 | 54.97 | 55.21 | 55.83 | 55.52 | 55.19 | 54.20 |

| $\gamma(\times 10^4)$ | 0.01 | 0.10 | 0.50 | 0.60 | 0.80 | 1.00 | 1.50 | 2.00 | 5.00 | 10.0 | 100.0 |
|---|---|---|---|---|---|---|---|---|---|---|---|
| worst-case (%) | 1.21 | 1.79 | 43.85 | 49.73 | 53.69 | 53.88 | 52.87 | 52.91 | 51.64 | 50.72 | 49.86 |
| average (%) | 50.61 | 50.78 | 52.21 | 52.83 | 53.49 | 55.59 | 54.17 | 54.93 | 53.31 | 51.35 | 50.17 |

Table 9: The performance of FedSDR on CFMNIST dataset with different hyper-parameters $\lambda$ and $\gamma$.

| $\lambda$ | 0.10 | 0.20 | 0.30 | 0.40 | 0.50 | 0.60 | 0.70 | 0.80 | 0.90 | 1.00 | 10.0 |
|---|---|---|---|---|---|---|---|---|---|---|---|
| worst-case (%) | 56.31 | 56.17 | 56.89 | 56.57 | 56.92 | 56.18 | 56.36 | 57.15 | 56.57 | 55.36 | 54.62 |
| average (%) | 61.62 | 62.07 | 61.78 | 61.57 | 61.88 | 61.83 | 61.75 | 61.53 | 61.25 | 61.03 | 60.77 |

| $\gamma(\times 10^4)$ | 0.01 | 0.10 | 0.50 | 0.60 | 0.80 | 1.00 | 1.50 | 2.00 | 5.00 | 10.0 | 100.0 |
|---|---|---|---|---|---|---|---|---|---|---|---|
| worst-case (%) | 0.19 | 0.27 | 52.68 | 55.57 | 56.28 | 56.92 | 54.79 | 53.15 | 53.05 | 51.68 | 49.59 |
| average (%) | 50.19 | 50.73 | 57.63 | 60.54 | 61.72 | 61.88 | 61.06 | 59.53 | 58.74 | 55.24 | 51.38 |

## C.6   COMPLETE EXPERIMENTAL RESULTS

The test accuracy shown in the main text are calculated based on 3 trials with different random seeds. Due to space limitation, we only give the average value in the evaluation part of the main text. Here, the complete results on CMNIST, CFMNIST and WaterBird are provided in Table 11 and the detailed results on PACS are shown in Table 12. Besides the detailed results for evaluating the effect of hyper-parameter $\lambda$ and $\gamma$ on WaterBird dataset is given in Table 10.

Table 10: The performance of FedSDR on WaterBird dataset with different hyper-parameters $\lambda$ and $\gamma$.

| $\lambda$ | 0.10 | 0.20 | 0.30 | 0.40 | 0.50 | 0.60 | 0.70 | 0.80 | 0.90 | 1.00 | 10.0 |
|---|---|---|---|---|---|---|---|---|---|---|---|
| worst-case (%) | 62.51 | 64.13 | 63.87 | 64.91 | 65.25 | 64.16 | 63.24 | 63.67 | 62.15 | 61.68 | 61.74 |
| average (%) | 72.39 | 71.87 | 71.56 | 72.78 | 73.20 | 72.15 | 71.76 | 72.09 | 71.14 | 70.61 | 70.18 |
| $\gamma$ | 0.10 | 1.00 | 1.10 | 1.20 | 1.30 | 1.40 | 1.50 | 1.60 | 1.80 | 2.00 | 10.0 |
| worst-case (%) | 44.16 | 57.64 | 60.33 | 61.58 | 63.75 | 65.25 | 64.52 | 63.26 | 63.58 | 60.95 | 48.29 |
| average (%) | 65.73 | 70.18 | 71.78 | 71.13 | 72.73 | 73.20 | 72.14 | 70.54 | 71.19 | 68.73 | 64.86 |

Table 11: The complete results on CMNIST, CFMNIST and WaterBird datasets.

| Dataset | CMNIST | | CFMNIST | | WaterBird | |
|---|---|---|---|---|---|---|
| Test case | worst-case (%) | average (%) | worst-case (%) | average (%) | worst-case (%) | average (%) |
| FedAvg | 3.39($\pm$0.29) | 51.03($\pm$0.67) | 0.16($\pm$0.36) | 50.02($\pm$0.42) | 54.13($\pm$0.75) | 67.95($\pm$1.03) |
| DRFA | 21.15($\pm$0.21) | 52.81($\pm$0.45) | 19.84($\pm$1.21) | 53.88($\pm$1.80) | 59.75($\pm$1.41) | 68.39($\pm$1.50) |
| FedSR | 46.93($\pm$0.12) | 48.62($\pm$0.53) | 47.61($\pm$0.39) | 48.90($\pm$0.53) | 61.75($\pm$0.81) | 71.68($\pm$1.17) |
| FedIIR | 47.25($\pm$0.26) | 48.39($\pm$0.42) | 48.06($\pm$0.50) | 49.16($\pm$0.51) | 61.24($\pm$0.76) | 70.87($\pm$0.94) |
| FTFA | 15.42($\pm$0.28) | 54.96($\pm$0.71) | 11.35($\pm$0.81) | 53.52($\pm$0.65) | 54.38($\pm$1.04) | 69.68($\pm$1.36) |
| pFedMe | 21.30($\pm$0.10) | 48.53($\pm$0.24) | 4.22($\pm$0.44) | 51.26($\pm$0.51) | 55.63($\pm$0.64) | 68.24($\pm$0.83) |
| Ditto | 3.02($\pm$0.32) | 50.97($\pm$0.59) | 0.37($\pm$0.42) | 50.12($\pm$0.63) | 53.13($\pm$0.53) | 68.73($\pm$0.72) |
| FedRep | 2.76($\pm$0.34) | 50.83($\pm$0.68) | 0.11($\pm$0.37) | 50.01($\pm$0.49) | 52.88($\pm$0.56) | 70.23($\pm$0.92) |
| FedRoD | 9.09($\pm$0.32) | 50.84($\pm$0.21) | 1.23($\pm$0.15) | 51.57($\pm$0.28) | 52.36($\pm$0.78) | 70.86($\pm$0.83) |
| FedPAC | 1.01($\pm$0.39) | 50.05($\pm$0.54) | 0.16($\pm$0.11) | 50.13($\pm$0.41) | 45.08($\pm$0.84) | 65.57($\pm$1.12) |
| **FedSDR** | **53.88** ($\pm$0.24) | **55.59** ($\pm$0.65) | **56.92** ($\pm$0.73) | **61.88** ($\pm$1.02) | **65.25** ($\pm$0.97) | **73.20** ($\pm$1.28) |

Table 12: The detailed results on PACS dataset.

| Algorithm | Art Painting (%) | Cartoon (%) | Photo (%) | Sketch (%) | average (%) |
|---|---|---|---|---|---|
| FedAvg | 41.71($\pm$0.62) | 51.21($\pm$0.16) | 44.79($\pm$0.27) | 52.92($\pm$0.44) | 47.66 |
| DRFA | 42.48($\pm$0.48) | 52.84($\pm$0.33) | 47.12($\pm$0.52) | 53.37($\pm$0.64) | 48.95 |
| FedSR | 46.76($\pm$0.34) | 54.15($\pm$0.24) | 48.67($\pm$0.43) | 55.43($\pm$0.71) | 51.25 |
| FedIIR | 47.03($\pm$0.40) | 54.66($\pm$0.34) | 49.51($\pm$0.49) | 55.12($\pm$0.85) | 51.58 |
| FTFA | 40.89($\pm$0.51) | 52.56($\pm$0.28) | 47.38($\pm$0.76) | 54.31($\pm$0.37) | 48.79 |
| pFedMe | 45.24($\pm$0.43) | 54.57($\pm$0.52) | 49.19($\pm$0.56) | 56.33($\pm$0.76) | 51.33 |
| Ditto | 44.95($\pm$0.57) | 53.48($\pm$0.21) | 50.71($\pm$0.65) | 55.98($\pm$0.57) | 51.28 |
| FedRep | 49.27($\pm$0.26) | 56.08($\pm$0.35) | 51.48($\pm$0.73) | 58.15($\pm$0.82) | 53.75 |
| FedRoD | 48.16($\pm$0.48) | 55.13($\pm$0.41) | 51.02($\pm$0.81) | 57.38($\pm$0.69) | 52.92 |
| FedPAC | 49.93($\pm$0.31) | 55.86($\pm$0.53) | 52.35($\pm$0.58) | 58.67($\pm$0.58) | 54.20 |
| IRM[†] | 46.35($\pm$0.81) | 53.71($\pm$0.79) | 48.52($\pm$0.86) | 54.73($\pm$0.97) | 50.83 |
| IRM[†]-FT | 43.18($\pm$0.56) | 53.23($\pm$0.45) | 48.27($\pm$0.66) | 55.48($\pm$0.73) | 50.04 |
| IRM[†]-L2 | 48.57($\pm$0.36) | 54.08($\pm$0.27) | 49.35($\pm$0.51) | 55.91($\pm$0.42) | 51.98 |
| **FedSDR** | 52.14($\pm$0.75) | 58.23($\pm$0.84) | 55.49($\pm$0.67) | 58.84($\pm$0.91) | 56.18 |
| FedSDR+FedAvg | 53.26($\pm$0.63) | 58.91($\pm$0.76) | 56.37($\pm$0.52) | 59.37($\pm$0.75) | 56.98 |
| FedSDR+pFedMe | 52.58($\pm$0.52) | 58.63($\pm$0.57) | 55.85($\pm$0.49) | 58.72($\pm$0.60) | 56.45 |

