# OpenReview forum: "Learning Personalized Causally Invariant Representations for Heterogeneous Federated Clients"
_ICLR.cc/2024/Conference — ICLR 2024 poster_

### Official Review · Reviewer_Qvuh · 2023-10-31

**Soundness:** 2 fair
**Presentation:** 2 fair
**Contribution:** 2 fair
**Rating:** 3
**Confidence:** 4

**Summary:**

The paper proposes a novel approach to mitigating shortcut learning in personalized federated learning, which is a challenging problem in real-world settings. The proposed method, FedSDR, utilizes structural causal models to discover and remove shortcuts while preserving personalization knowledge.

**Strengths:**

1. This paper provides extensive background information, offering readers significant convenience in understanding the topic.

2. The authors conducted extensive experiments on some real-world datasets, validating the excellent performance of the proposed method.

**Weaknesses:**

1. I believe the main issue with this paper is that the research motivation seems weak. The paper claims to primarily address the scenario where training sets are Non-IID among clients, and where within each client, the training and test sets are also Non-IID. However, I think existing Robust Federated Learning and Federated Domain Generalization methods are capable of handling the aforementioned scenario. Although these two methods primarily focus on the issue of Non-IID training and test sets within each client.

2. The paper states, "To the best of our knowledge, we are the first to consider the shortcut trap problem in personalized federated learning and analyze it by formulating the structural causal models for heterogeneous clients." While it's true that this paper is indeed the first to study the use of PFL+SCM to address the shortcut trap problem, existing Robust Federated Learning and Federated Domain Generalization methods can also address the shortcut trap problem. Therefore, the contribution of this paper appears to be limited.

3. In Figure 1, I find the classification of the scenarios where RFL&FedDG are applicable not very accurate. In the scenarios where RFL&FedDG are applicable, the Test-test relation can also be IID or Non-IID.

4. In the experimental section, the proposed model is only compared to FL and PFL methods. However, I believe it should also be compared to RFL and FedDG methods to provide more convincing experimental results.

5. The experimental section lacks detailed information about the configuration of the model used in this paper. More detailed information is needed.

6. The paper lacks a thorough discussion of the limitations of the proposed FedSDR method. While the authors mention some potential limitations in passing, a more detailed discussion of the assumptions and constraints of the method would be helpful for readers to better understand its applicability in different scenarios.

7. Although the authors provide some high-level explanations of how FedSDR works, a more detailed discussion of the causal modeling framework and how it is used to address shortcut learning would be helpful for readers who are not familiar with this area of research.

**Questions:**

Please see above

---

> ### Author Response · Authors · 2023-11-22
> **Responses to Reviewer Qvuh**
>
> Thanks for the reviewer's valuable feedbacks and suggestions. The detailed responses to the reviewer's questions and concerns are listed as follows:
>
> __1. I believe the main issue with this paper is that the research motivation seems weak. The paper claims to primarily address the scenario where training sets are Non-IID among clients, and where within each client, the training and test sets are also Non-IID. However, I think existing Robust Federated Learning and Federated Domain Generalization methods are capable of handling the aforementioned scenario. Although these two methods primarily focus on the issue of Non-IID training and test sets within each client.__
>
> __Answer:__ Unfortunately, both robust federated learning and domain generalization methods cannot handle the scenarios well where target datasets are Non-IID across local clients. Since both Federated Domain Generalization (FedDG) and Robust Federated Learning (RFL) generate a shared global model, __they abandon the important personalization information on each client.__ Therefore, the relation between our method and RFL/FedDG is same as the relation between the existing personalized federated learning (FL) and traditional federated learning (FL).
>
> __2. The existing Robust Federated Learning and Federated Domain Generalization methods can also address the shortcut trap problem. Therefore, the contribution of this paper appears to be limited.__
>
> __Answer:__ The existing Robust Federated Learning (RFL) and Federated Domain Generalization (FedDG) methods will abandon the important personalization information on each client since they output a shared global model, although they can also mitigate the shortcut features. In many real-world applications (e.g., medical diagnosis and automatic driving), personalization information is of vital importance and cannot be overlooked. The main contribution of this paper is to fully exploit the significant personalization information, and in the meanwhile eliminate the shortcut features. It's rather challenging because the personalized and shortcut features are usually entangled in the real-world scenarios.
>
> __3. In Figure 1, I find the classification of the scenarios where RFL&FedDG are applicable not very accurate. In the scenarios where RFL&FedDG are applicable, the Test-test relation can also be IID or Non-IID.__
>
> __Answer:__ The existing Robust Federated Learning (RFL) and Federated Domain Generalization (FedDG) methods will abandon the important personalization information on each client since they output a shared global model, when target datasets are Non-IID across local clients. Therefore, we conclude that RFL&FedDG can't cover the scenario well. The relation between our method and RFL/FedDG is same as the relation between the existing personalized federated learning (FL) and traditional federated learning (FL).
>
> __4. In the experimental section, the proposed model is only compared to FL and PFL methods. However, I believe it should also be compared to RFL and FedDG methods to provide more convincing experimental results.__
>
> __Answer:__ Actually, __we included three RFL&FedDG methods as baselines in the experimental section.__ The considered competitor DRFA is a typical RFA method while FedSR and FedIIR are two state-of-the-art FedDG methods. The overall experimental results of these baseline methods can be found in Table 1.
>
> [1] Yuyang Deng, Mohammad Mahdi Kamani, and Mehrdad Mahdavi. Distributionally robust federated averaging. Advances in Neural Information Processing Systems, 33:15111–15122, 2020.
>
> [2] A Tuan Nguyen, Philip Torr, and Ser-Nam Lim. Fedsr: A simple and effective domain generalization method for federated learning. In Advances in Neural Information Processing Systems, 2022.
>
> [3] Yaming Guo, Kai Guo, Xiaofeng Cao, Tieru Wu, and Yi Chang. Out-of-distribution generalization of federated learning via implicit invariant relationships. In Proceedings of the 40th International Conference on Machine Learning, pp. 11905–11933, 2023.
>
> __5. The experimental section lacks detailed information about the configuration of the model used in this paper. More detailed information is needed.__
>
> __Answer:__ The detailed information about the adopted models were provided in section 5.1. For CMNIST and CFMNIST, we adopt the deep neural network with one hidden layer of (128, 128) as feature extractor and an subsequent fully-connected layer as classifier. As regard to Waterbird and PACS, ResNet-18 is used as the learning model where the part before the last fully-connected layer works as feature extractor and the last fully connected layer works as classifier.

---

> > ### Author Response · Authors · 2023-11-22
> > **Responses to Reviewer Qvuh**
> >
> > __6. The paper lacks a thorough discussion of the limitations of the proposed FedSDR method. While the authors mention some potential limitations in passing, a more detailed discussion of the assumptions and constraints of the method would be helpful for readers to better understand its applicability in different scenarios.__
> >
> > __Answer:__ We believe the outstanding limitation of this work is that the provided theoretical guarantee is restricted to linear feature space and we mentioned it in conclusion section. We will include more discussions in the updated version. Of course, extending the theoretical guarantee to more complex cases is rather challenging and we will push ahead with it in the future work.
> >
> > __7. Although the authors provide some high-level explanations of how FedSDR works, a more detailed discussion of the causal modeling framework and how it is used to address shortcut learning would be helpful for readers who are not familiar with this area of research.__
> >
> > __Answer:__ Actually, a reasonable structural causal model can be regarded as some prior knowledge. We propose the structural causal models in federated learning by adding the user/client indicator $U$ and deconstructing the invariant features into two separate parts: the personalized invariance $Z_C^U$ and the shared/global invariance $Z_C^g$. Other parts of the causal models are same as what are commonly adopted in the literature on centralized invariant learning.
> >
> > As for how they are used to address shortcut trap in federated learning, the key is two provable causal signatures that we observe from the proposed structural causal models. As listed in Lemma 1, the first causal signature indicates that we can discover the shortcut features using training environments across local clients even if the data generating mechanisms are heterogeneous among them. The second causal signature makes it possible to develop the optimal personalized invariant predictors with the discovered shortcut features even though there is just one training environment on each client, since the relationships between $Z_C^g$, $Z_C^U$ and $Z_S$ are independent of environment $E$. With these two causal signatures, we design the provable shortcut discovery and removal method for federated learning, i.e., FedSDR.
> >
> > __Thanks again for your valuable feedbacks. More discussions are welcomed if you still have any concerns or questions for this work.__

---

> ### Author Response · Authors · 2023-11-23
>
> Dear Reviewer Qvuh,
>
> Thanks for your valuable feedbacks. As the deadline for author-reviewer discussion is approaching, we are concerned you might miss the clarifications that we make for your questions. We would appreciate it very much if you could reassess our work according to the clarifications. Of course, more discussions are welcomed if you still have any concerns or questions for this work.

---

### Official Review · Reviewer_2uxd · 2023-11-01

**Soundness:** 3 good
**Presentation:** 3 good
**Contribution:** 3 good
**Rating:** 6
**Confidence:** 3

**Summary:**

The paper proposes a new personalized federated learning approach named FedSDR. Considering generalization on unseen test data, the paper utilizes invariant learning in the federated setting. Specifically, FedSDR first extracts shortcut features that are irrelevant to the task and them remove it to extract the most informative personalized invariant features by carefully designing the objectives. Experiments show that FedSDR outperforms the other baselines in the settings where the test data distribution shifts.

**Strengths:**

1. Applying invariant learning in the federated setting is interesting and promising.

2. The organization of the paper is clear.

3. FedSDR significantly outperforms the other baselines.

**Weaknesses:**

1. My main concern is about the experimental setting. Experiments are based on the simulated setting where the authors manually add shortcut features and change the test data distributions. Based on the motivation, the shortcut features should naturally exist in datasets. Experiments on real-world natural datasets are necessary. Otherwise, the application of FedSDR may be very limited.

2. The theoretical analysis has strong assumptions, e.g., logistic regression, linear function, etc. The analysis may not be applicable in the experimental settings.

3. The number of clients used in the experiments is missing. Experiments to evaluate the scalability of FedSDR are not provided.

4. Typos: 1. Page 4: “Theorem 1” -> “Lemma 1”; 2. Page 6: “guarantee” -> “guarantees”

**Questions:**

1. Can you add a synthetic dataset with a simple model in the experiments? It can be used to verify the theorems by satisfying the assumptions.

2. Can you add experiments without manually adding shortcut features? It is quite important. Currently, I’m not clear what are the real applications of FedSDR.

3. Can you add experiments that increase the number of clients and adopt client sampling?

---

> ### Author Response · Authors · 2023-11-22
> **Part 1 of responses to Reviewer 2uxd**
>
> Thanks a lot for the reviewer's valuable feedbacks and suggestions. The detailed responses to the reviewer's questions and concerns are listed as follows:
>
> __1. Can you add a synthetic dataset with a simple model in the experiments? It can be used to verify the theorems by satisfying the assumptions.__
>
> __Answer:__ We generate a synthetic dataset using the same strategy as in [2]. Specifically, it is a logistic regression task and the data instance $X$ is generated by $X=g(Z_C^g, Z_C^U, Z_S)$, where the dimensionalities of $Z_C^g$, $Z_C^U$ and $Z_S$ are $d_C^g=3$, $d_C^U=3$ and $d_S=6$ respectively. The linear function $g$ is implemented by one fully-connected layer which has 12 neurons. The latent variables $Z_C^g$, $Z_C^U$ and $Z_S$ are subject to $\mathcal{N}(y\cdot \mu_{c,g}, \sigma_{c,g}^2I)$, $\mathcal{N}(y\cdot \mu_{c,u}, \sigma_{c,u}^2I)$ and $\mathcal{N}(y\cdot \mu_s, \sigma_s^2I)$ respectively. Target variable $y$ is taken from the distribution $\mathbb{P}(y=-1)=\mathbb{P}(y=1)=0.5$. Both $\mu_{c,g}$ and $\mu_{c,u}$ are randomly sampled from $\mathcal{N}(0, 1.5I)$ while $u_s$ is randomly sampled from $\mathcal{N}(0, 0.75I)$. To make the shortcut representation $Z_S$ easier to learn, we choose $\sigma_{c,g}=\sigma_{c,u}=2$ and $\sigma_s=1$ as in [2]. Each fixed value of $\mu_s$ indicates one environment. We generate __10__ training environments and __5000__ test environments to evaluate the out-of-distribution generalization performance. Each (training/test) environment contains 10000 data samples (X, y) and the training data samples are distributed onto totally __100__ clients. The training and test data samples on each client are generated with an identical value of $\mu_{c,u}$. Besides, we choose the client sampling rate as 0.1.  The experimental results on this synthetic dataset are shown in the following table:
>
> | Algorithm | FedAvg |  DRFA   | FedSR   | FedIIR  | FTFA   | pFedMe  |  Ditto  | FedRep  | FedRoD  | FedPAC  | FedSDR |
> | :---  | :---: | :---: | :---: | :---: | :---: | :---: | :---: | :---: | :---: | :---: | :---: |
> | worst-case (\%)| $3.06$  | $62.41$ | $63.09$ | $67.39$ |  $1.32$  | $10.58$ |  $7.76$  |  $2.57$  | $21.53$ | $8.98$  | $92.49$ |
> | average     (\%)| $85.56$ | $69.64$ | $70.53$ | $70.75$ | $96.26$ | $95.72$ | $96.50$ | $97.80$ | $97.24$ | $98.77$ | $96.07$ |
>
> In particular, when we manually select the causal features $[Z_C^g, Z_C^U]$ as the discriminating features, we find the optimal personalized classifiers achieve an stable accuracy around $97.5$ in different test environments. Therefore, the results shown in Table 1 can demonstrate the effectiveness of our FedSDR on developing the optimal personalized invariant predictors, compared with the state-of-the-art FL and PFL methods.
>
> [2] Rosenfeld, E., Ravikumar, P., & Risteski, A. (2021, July). The Risks of Invariant Risk Minimization. In International Conference on Learning Representations.
>
> __2. Can you add experiments without manually adding shortcut features? It is quite important. Currently, I’m not clear what are the real applications of FedSDR.__
>
> __Answer:__ Actually, __PACS__ is a real-world dataset that is commonly used in related papers. It's a multi-class classification task and there doesn't exists explicit (or manually adding) shortcut features. Specifically, It consists of 7 classes distributed across 4 environments (or domains). The results on PACS dataset are also shown in Table 1.

---

> > ### Author Response · Authors · 2023-11-22
> > **Part 2 of responses to Reviewer 2uxd**
> >
> > __3. The number of clients used in the experiments is missing. Experiments to evaluate the scalability of FedSDR are not provided. Can you add experiments that increase the number of clients and adopt client sampling?__
> >
> > __Answer:__ Due to the space limitation, we put the complete experimental setups in Appendix B.1. In particular, we simulate 8 clients in the experiments on CMNIST, CFMNIST and WaterBird. The experiments on PACS are conducted on 6 clients. To further evaluate the scalability of FedSDR, we firstly partition CMNIST dateset into 8 subsets using the same strategy adopted to simulate 8 clients. And then  , we randomly distribute each subset onto 10 clients. In this way, we totally construct 80 clients for CMNIST dataset. Similarly, we construct 80, 80, 60 clients for CFMNIST, WaterBird and PACS respectively. When evaluating the model performance on these four datasets, we adopt __a client sampling rate of 0.1__. The experimental results are shown in the following table:
> >
> > | Dataset | CMNIST | | CFMNIST | | WaterBird | | PACS | |
> > | :--- | :---: | :---: | :---: | :---: | :---: | :---: | :---: | :---: |
> > | Test accuracy (\%) | worst-case | average | worst-case | average | worst-case | average | worst-case | average |
> > | FedAvg    | $1.74$  | $46.82$ | $0.77$  | $45.62$ | $48.65$ | $61.57$ | $33.75$ | $40.18$ |
> > | DRFA      | $14.94$ | $47.24$ | $15.51$ | $47.14$ | $52.34$ | $60.43$ | $36.17$ | $41.75$ |
> > | FedSR    | $40.29$ | $43.64$ | $41.16$ | $43.27$ | $\underline{55.63}$ | $64.32$ | $39.03$ | $43.40$ |
> > | FedIIR    | $\underline{41.18}$ | $42.93$ | $\underline{41.80}$ | $43.58$ | $54.31$ | $64.60$ | $40.15$ | $44.37$ |
> > | FTFA      | $11.51$ | $\underline{49.28}$ | $7.20$ | $47.57$ | $50.25$ | $63.39$ | $34.65$ | $42.19$ |
> > | pFedMe | $17.28$ | $44.13$ | $2.42$  | $\underline{47.95}$ | $50.01$ | $61.97$ | $41.06$ | $45.84$ |
> > | Ditto       | $1.98$  | $45.84$ | $1.80$  | $45.71$ | $49.08$ | $63.38$ | $40.18$ | $46.30$ |
> > | FedRep  | $1.56$  | $46.20$ | $0.83$  | $46.14$ | $48.12$ | $64.52$ | $42.16$ | $47.58$ |
> > | FedRoD  | $6.53$  | $46.86$ | $1.60$ | $47.43$ | $49.56$ | $\underline{65.49}$ | $42.68$ | $46.61$  |
> > | FedPAC  | $0.38$  | $45.64$ | $0.23$  | $44.88$ | $42.61$ | $63.81$ | $\underline{44.19}$ | $\underline{49.71}$ |
> > |__FedSDR__| __50.41__ | __51.85__ | __52.81__ | __57.14__ | __59.96__ | __68.09__ | __48.07__ | __51.55__ |
> >
> > The results show that FedSDR can still outperform the competitors when there are a large number of clients in the federated learning system, which can validate the scalability of the proposed FedSDR.
> >
> > __4. The theoretical analysis has strong assumptions, e.g., logistic regression, linear function, etc. The analysis may not be applicable in the experimental settings.__
> >
> > __Answer:__ Actually, the algorithm itself doesn't require the assumption of linear feature space. In particular, the experiments on WaterBird and PACS are conducted with ResNet as learning model. Therefore, feature spaces of these two datasets are non-linear with regard to input $X$. Moreover, PACS is a multi-class classification task rather than logistic regression task. The results on thses two datasets can demonstrate the effectiveness of FedSDR in general scenarios (including non-linear cases and multi-class classification).
> >
> > Of course, the theoretical guarantee is still restricted to linear cases in the current version. Extending the theoretical guarantee to more complex cases is rather challenging and we will push ahead with it in the future work.
> >
> > __5. Typos: 1. Page 4: “Theorem 1” -> “Lemma 1”; 2. Page 6: “guarantee” -> “guarantees”__
> >
> > __Answer:__ We will fix them in the updated version.
> >
> > Thanks again for your valuable feedbacks. More discussions are welcomed if you still have any concerns or questions for this work.

---

> > > ### Comment · Reviewer_2uxd · 2023-11-23
> > >
> > > Thanks for your response. Most of my concerns have been addressed. I have raised my score to 6.

---

> > > > ### Author Response · Authors · 2023-11-23
> > > > **Responses to Reviewer 2uxd**
> > > >
> > > > Thanks a lot for your reassessment of our work. Thanks again for your valuable feedbacks which help to improve the completeness of this work.

---

### Official Review · Reviewer_zjFS · 2023-11-03

**Soundness:** 3 good
**Presentation:** 3 good
**Contribution:** 3 good
**Rating:** 6
**Confidence:** 4

**Summary:**

The paper focuses on addressing the "shortcut trap" issue within personalized federated learning (pFL) by introducing FedSDR. This solution aims to identify and eliminate shortcut features, leading to enhanced performance for individual clients within pFL on their respective local datasets. The authors support their approach with theoretical proofs and comprehensive experiments, demonstrating the effectiveness of FedSDR across multiple scenarios.

**Strengths:**

1. Originality: The paper pioneers the exploration of the shortcut trap problem in pFL.

2. Solid Theoretical Support: The inclusion of strong theoretical foundations.

3. Sufficient Experiments: The comprehensive set of experiments strengthens the paper's contributions.

**Weaknesses:**

1. Notation Conciseness: The notation could be more concise, potentially enhancing the paper's readability.

2. Confusing Training Process: A section of the training process appears confusing and requires clarification.

3. Inconclusive Experimental Results: Certain aspects of the experimental outcomes lack conviction and need further clarification for robustness.

**Questions:**

1. The paper’s algorithmic approach (Algorithm 1) deviates from the conventional practice of client selection in Federated Learning (FL). Could the authors elucidate why they've chosen this approach over the typical FL client selection methodology?

2. Concerns arise from certain experimental results in Table 1, particularly the unexpected lower accuracies observed for specific baseline methods on CMNIST and CFMNIST. Can the authors provide an explanation to address these concerns?

---

> ### Author Response · Authors · 2023-11-22
> **Responses to Reviewer zjFS**
>
> Thanks a lot for the reviewer's valuable comments and suggestions. The detailed responses to the reviewer's questions and concerns are listed as follows:
>
> __1. The paper’s algorithmic approach (Algorithm 1) deviates from the conventional practice of client selection in Federated Learning (FL). Could the authors elucidate why they've chosen this approach over the typical FL client selection methodology?__
>
> __Answer:__ We adopted the same federated learning framework as in pFedMe [1] while the conventional client selection is conducted before local updates. In the adopted framework, the global model is generated by aggregating the updates from the selected clients and the updates from the clients that are not selected are never used. Then on each client, the local model will be initialized by the global model before local update. __Therefore, the final output models of these two client selection methods are exactly identical.__ Our method FedSDR can also be implemented using the conventional client selection methodology and the experimental results will keep unchanged.
>
> [1] T Dinh, C., Tran, N., & Nguyen, J. (2020). Personalized federated learning with moreau envelopes. Advances in Neural Information Processing Systems, 33, 21394-21405.
>
> __2. Concerns arise from certain experimental results in Table 1, particularly the unexpected lower accuracies observed for specific baseline methods on CMNIST and CFMNIST. Can the authors provide an explanation to address these concerns?__
>
> __Answer:__ As shown in Table 1, our FedSDR achieved around __6.5\%__, __9\%__, __3.5\%__ and __2\%__ higher worst-case accuracy than the second best algorithm among the baselines and in the meanwhile reaches the highest average accuracy on CMNIST, CFMNIST, WaterBird and PACS, respectively. So, what do you mean by "the unexpected lower accuracies observed for specific baseline methods on CMNIST and CFMNIST"? __In addition, we will make the project codes public soon to enhance the reproducibility of the conducted experiments.__
>
> Thanks again for your valuable feedbacks. More discussions are welcomed if you still have any concerns or questions for this work.

---

> > ### Comment · Reviewer_zjFS · 2023-11-22
> >
> > Thanks a lot for your response. I would like to inquire further about the reasons behind the significantly low worst-case results obtained by pFL baseline methods. It would be greatly appreciated if the authors could provide a reasonable explanation.

---

> > > ### Author Response · Authors · 2023-11-22
> > > **About the low worst-case results obtained by pFL baseline methods.**
> > >
> > > Glad to receive your further responses.
> > >
> > > Good question. The prevalent PFL methods develop the personalized model on each client with the guidance of a shared global model (e.g., FTFA, pFedMe and Ditto) or some shared knowledge (e.g., FedRep, FedRoD and FedPAC). Since both personalization information and environment-dependent shortcut features are varied across local clients, the shared global model/knowledge excludes both personalized and shortcut features. When these methods conduct local adaptation to develop the personalized model on each client, the developed model can readily pick up the shortcut features if local training data is biased because these PFL methods lack a strong mechanism for shortcut identification and removal. However, relying on shortcut features can degrade the performance of the developed model to any extent when the distribution shift between training and test datasets is very large.  In other words, the worst-case accuracy can be  significantly low. In comparison, the proposed FedSDR can effectively extract the shortcut features and then eliminate the identified shortcut features during the local adaptation. Therefore, FedSDR can achieve a much stabler and higher test accuracy than the PFL baseline methods.
> > >
> > > Thanks again for your further responses. More discussions are welcomed if you still have any concerns or questions for this work.

---

### Meta-Review · Area_Chair_GVQR · 2023-12-06

**Metareview:**

This paper introduces FedSDR, a method for addressing shortcut learning in personalized federated learning (PFL). PFL faces challenges in real-world settings due to the assumption of in-distribution test data, leading to shortcut learning. FedSDR utilizes structural causal models for federated clients, offering provable shortcut discovery and removal. It collaboratively discovers shortcuts using training data and develops an optimal personalized causally invariant predictor for each client. Theoretical analysis and experiments on diverse datasets validate FedSDR's superiority over state-of-the-art PFL methods in out-of-distribution generalization performance.

The paper got controversial comments from the reviewers. While most of the reviewers recognize the technical novelty and effectiveness of this paper, one reviewer has some concern on the comparison with robust federated learning and federated domain generalization. As discussed in the rebuttal, the reviewer ignored such discussion and comparison in the paper. Therefore, despite the overall low rating, mainly influenced by one significantly low score, I lean towards recommending the acceptance of the paper.

**Justification For Why Not Higher Score:**

overall rating is not high

**Justification For Why Not Lower Score:**

The paper makes a good contribution to causal treatment of federated learning. The technical quality is good and the experiments are solid.

As the overall score is low, due to a bias in one reviewer, I am open to discussion if there are some concerns.

---

### Decision · Program_Chairs · 2024-01-16

Accept (poster)